# Bed nucleus of the stria terminalis regulates fear to unpredictable threat signals

**Travis D Goode, Reed L Ressler, Gillian M Acca, Olivia W Miles, Stephen Maren***

Department of Psychological and Brain Sciences, Institute for Neuroscience, Texas A&M University, College Station, United States

**Abstract** The bed nucleus of the stria terminalis (BNST) has been implicated in conditioned fear and anxiety, but the specific factors that engage the BNST in defensive behaviors are unclear. Here we examined whether the BNST mediates freezing to conditioned stimuli (CSs) that poorly predict the onset of aversive unconditioned stimuli (USs) in rats. Reversible inactivation of the BNST selectively reduced freezing to CSs that poorly signaled US onset (e.g., a backward CS that followed the US), but did not eliminate freezing to forward CSs even when they predicted USs of variable intensity. Additionally, backward (but not forward) CSs selectively increased Fos in the ventral BNST and in BNST-projecting neurons in the infralimbic region of the medial prefrontal cortex (mPFC), but not in the hippocampus or amygdala. These data reveal that BNST circuits regulate fear to unpredictable threats, which may be critical to the etiology and expression of anxiety.

DOI: https://doi.org/10.7554/eLife.46525.001

*For correspondence:
maren@tamu.edu

**Competing interests:** The authors declare that no competing interests exist.

## Introduction

Excessive apprehension about potential future threats, including financial loss, illness, or death, is a defining feature of many anxiety disorders, including generalized anxiety disorder (GAD) (*Behar et al., 2009*). Anxiety and trauma-related disorders are widespread and costly (*Blanco et al., 2011*; *Comer et al., 2011*; *Kinley et al., 2011*; *Salas-Wright et al., 2014*; *Stein et al., 2017*; *Wittchen, 2002*), and remain difficult to treat (*Colvonen et al., 2017*; *Costello et al., 2014*; *Iza et al., 2013*; *Sinnema et al., 2015*). Understanding the neural circuits underlying anxiety is important for refining behavioral and pharmacotherapeutic treatments (*Berridge, 2018*; *Deslauriers et al., 2018*; *Fanselow and Pennington, 2018*; *Fanselow and Pennington, 2017*; *Graham et al., 2014*; *LeDoux and Daw, 2018*; *Nees et al., 2015*; *Pine and LeDoux, 2017*; *Tye, 2018*). Several recent studies have demonstrated changes in the function of the bed nucleus of the stria terminalis (BNST) in individuals with post-traumatic stress disorder (PTSD) and anxiety disorders (*Andreescu et al., 2015*; *Brinkmann et al., 2018*; *Brinkmann et al., 2017b*; *Brinkmann et al., 2017a*; *Buff et al., 2017*; *Münsterkötter et al., 2015*; *Rabellino et al., 2018*; *Straube et al., 2007*; *Yassa et al., 2012*). However, the conditions that recruit the BNST to aversive learning and memory processes believed to underlie anxiety disorders are still not understood (*Avery et al., 2016*; *Avery et al., 2014*; *Ch'ng et al., 2018*; *Daniel and Rainnie, 2016*; *Davis et al., 2010*; *Fox and Shackman, 2019*; *Goode and Maren, 2017*; *Gungor and Paré, 2016*; *Lebow and Chen, 2016*; *Perusini and Fanselow, 2015*; *Shackman and Fox, 2016*; *Walker et al., 2009*; *Walker and Davis, 2008*).

Early work on this question revealed that BNST lesions in rats impair defensive behaviors evoked by unconditioned threats (*Gewirtz et al., 1998*). For example, BNST inactivation results in a loss of unconditioned defensive responding to the presence of predator odors (*Breitfeld et al., 2015*;

*Fendt et al., 2003*; *Xu et al., 2012*). Additionally, unconditioned increases in the acoustic startle reflex produced by either intracranial administration of corticotropin-releasing factor (CRF) or exposure to bright light require the BNST, whereas fear-potentiated startle to punctate conditioned stimuli (CSs) do not (*Walker et al., 2009*). However, the involvement of the BNST in defensive responding is not restricted to unconditioned threat: BNST lesions produce deficits in both freezing and corticosterone release elicited by contextual, but not auditory, CSs after Pavlovian fear conditioning in rats (*LeDoux et al., 1988*; *Sullivan et al., 2004*). Interestingly, however, it is not stimulus modality that differentiates the effects of BNST lesions on tone and context freezing. BNST lesions impair freezing responses to long-duration (e.g., 10 min) auditory conditioned stimuli (CSs) (*Waddell et al., 2006*), and spare freezing to short-duration (1 min) contextual stimuli (*Hammack et al., 2015*). Based on this work, it has been proposed that the BNST is required to organize behavioral and hormonal responses to sustained threats (whether conditioned or unconditioned) (*Hammack et al., 2015*; *Hammack et al., 2009*; *Takahashi, 2014*; *Waddell et al., 2008*; *Waddell et al., 2006*). Of course, the duration of the behavioral responses in these situations is confounded with the duration of the eliciting stimulus. Thus, it has been argued that the BNST mediates sustained defensive *responses* independent of the specific features and durations of the stimuli that evoke them (*Davis, 2006*; *Davis et al., 2010*; *Walker et al., 2009*; *Walker and Davis, 2008*).

Of course, another factor that differentiates discrete tones from contexts, or short- from long-duration stimuli (whether tones or contexts), is the temporal information the stimuli provide about when an aversive event will occur (*Goode and Maren, 2017*). For example, long-duration CSs or contexts provide relatively poor information about *when* a future unconditioned stimulus (US) will occur (i.e., the US will occur at some distal time in the future), whereas short-duration CSs or contexts provide more immediate certainty about the temporal imminence of US onset (i.e., the US will occur soon). We and others have proposed that the BNST may have a critical role in processing temporally unpredictable threats, particularly in mediating defensive responses to threatening stimuli that poorly predict when an aversive event will occur (*Goode and Maren, 2017*; *Lange et al., 2017*; *Luyck et al., 2018b*). Consistent with this view, punctate auditory CSs that are followed by shock at unpredictable latencies yield freezing responses that are sensitive to BNST manipulations (*Daldrup et al., 2016*; *Lange et al., 2017*). This suggests that a crucial parameter that determines the role for the BNST in defensive behavior is neither the duration nor modality of the threat (nor the duration of the elicited defensive response), but rather the information a signal provides about when an aversive event will occur.

Here we directly examined this possibility by investigating the role of the BNST in conditioned freezing responses elicited by procedures that equated both the duration and modality of the threat CSs, as well as the total number of US presentations, but differed according to the timing of the aversive US in relation to the CS. Specifically, we arranged a brief (10 s) auditory CS to either precede (forward conditioning, FW) or follow (backward conditioning, BW) a footshock US in rats. Although extensively-trained BW CSs become conditioned inhibitors that dampen responding to other first-order excitatory cues (*Andreatta et al., 2012*; *Ayres et al., 1976*; *Christianson et al., 2011*; *Gerber et al., 2014*; *Moscovitch and LoLordo, 1968*; *Siegel and Domjan, 1971*), minimally-trained BW CSs elicit excitatory conditioned responses that transfer across contexts (*Ayres et al., 1987*; *Barnet and Miller, 1996*; *Bevins and Ayres, 1992*; *Chang et al., 2003*; *Connor et al., 2017*; *Heth, 1976*; *Mahoney and Ayres, 1976*; *Prével et al., 2018*; *Prével et al., 2016*; *Rescorla, 1968*). After conditioning, we examined the effect of pharmacological inactivation of the BNST on freezing to FW or BW CSs. We hypothesized that pharmacological inactivation of the BNST would selectively disrupt fear expression to the BW CS and that the presentation of the BW CS would recruit BNST neurons (as assessed by Fos) in greater numbers than the FW CS. Furthermore, we anticipated that the backward CS would increase the activity of BNST-projecting neurons in afferent brain regions implicated in anxiety states. Consistent with our hypothesis, we found that BNST inactivation reduced freezing to a BW, but not FW, CS. This effect was related to the uncertainty of the BW CS in predicting US onset, because the same outcome was observed when USs occurred randomly with respect to the CS. We also observed that BW CSs selectively increased Fos expression in the BNST and BNST-projecting medial prefrontal cortical (mPFC) neurons. These data reveal that BNST circuits process the expression of defensive behavior in the presence of unpredictable threat signals.

## Results

### Reversible inactivation of the BNST attenuates fear to backward, not forward, CSs

To examine the role of the BNST in threat uncertainty, we reversibly inactivated the BNST during retrieval of fear to either a forward- ('FW'; predictable threat) or backward-trained ('BW'; unpredictable threat) CS. A schematic of the behavioral design is shown in *Figure 1A*. Representative cannula tracts and histological placements are presented in *Figure 1—figure supplement 1* and *Figure 1—figure supplement 2* (respectively). Freezing behavior during the conditioning session is shown in *Figure 1B*. A main effect of conditioning trial was observed (repeated measures: $F_{6,288} = 37.87$, $p < 0.0001$). No other significant main effects or interactions were observed for any of the conditioning or drug assignments across the conditioning trials ($F$'s < 0.70, $p$'s > 0.60).

One day after conditioning, the animals were infused with NBQX, an AMPA receptor antagonist, to reversibly inactivate the BNST; saline ('VEH') infusions served as a control. Immediately after the infusions, the rats were placed in a novel context and received twelve presentations of the CS [some BW-trained rats received no CS exposure at the test, 'NoCS(Test)"]. As shown in (*Figure 1B*), inactivation of the BNST produced a selective decrease in conditioned freezing to the BW CS. Analysis of freezing behavior across the entire session (including the baseline) revealed a main effect of trial

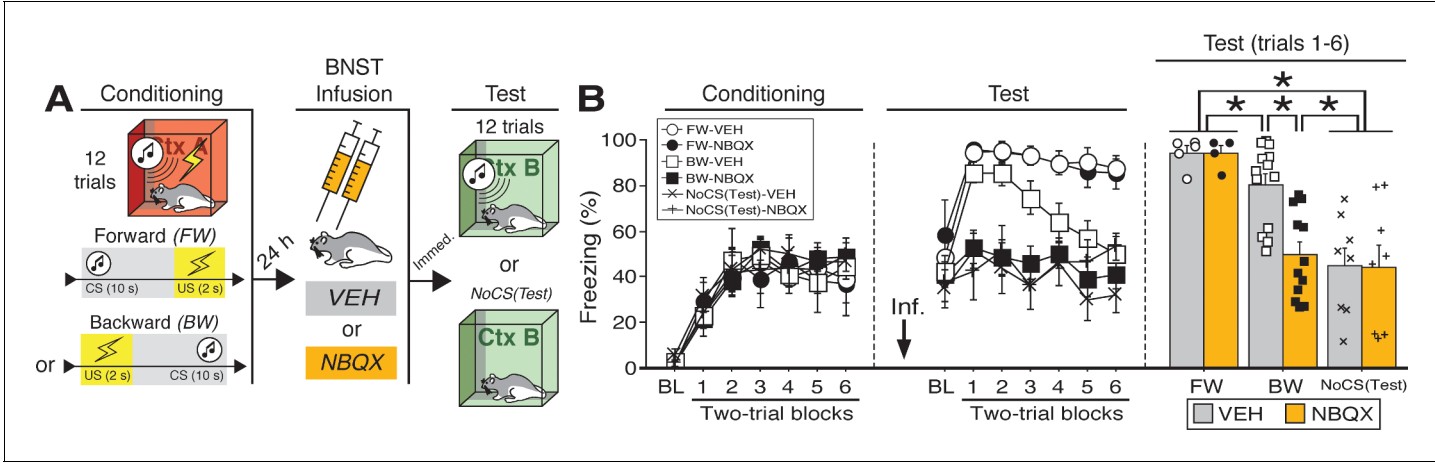

**Figure 1.** Reversible inactivation of the BNST attenuates conditioned fear expression to a backward, but not forward, CS. (**A**) Behavioral schematic. (**B**) Freezing behavior at conditioning and retrieval testing. For conditioning, the left panel depicts mean percentage freezing during the 5 min baseline (BL) and across each conditioning block (each 136 s block is comprised of two trials; conditioning trials consist of freezing during the 10 s CS followed by the 58 s interstimulus interval). For retrieval testing, the center panel shows mean percentage freezing at the 5 min baseline (BL) and across each test block (each 140 s block is comprised of two trials; trials consist of freezing during the 10 s CS followed by the 60 s interstimulus interval). The right panel shows mean percentage freezing during the first half of the test (trials 1–6; corresponding to 420 s of behavior). All data are represented as means ± s. e.m [FW-VEH (n = 5); FW-NBQX (n = 4); BW-VEH (n = 13); BW-NBQX (n = 12); NoCS(Test)-VEH (n = 8); NoCS(Test)-NBQX (n = 8)]; * = p < 0.05.

DOI: https://doi.org/10.7554/eLife.46525.002

The following source data and figure supplements are available for figure 1:

**Source data 1.**
DOI: https://doi.org/10.7554/eLife.46525.008

**Figure supplement 1.** Representative bilateral cannula placements in the BNST.
DOI: https://doi.org/10.7554/eLife.46525.003

**Figure supplement 2.** Bilateral cannula placements for BNST microinfusions.
DOI: https://doi.org/10.7554/eLife.46525.004

**Figure supplement 3.** Effects of BNST inactivation on freezing to a forward vs. backward CS trained with five trials.
DOI: https://doi.org/10.7554/eLife.46525.005

**Figure supplement 4.** Bilateral cannula placements for BNST microinfusions.
DOI: https://doi.org/10.7554/eLife.46525.006

**Figure supplement 5.** Shock-induced activity during conditioning to a forward vs. backward CS.
DOI: https://doi.org/10.7554/eLife.46525.007

(repeated measures: $F_{6,264}$ = 8.06, p < 0.0001), a significant main effect of CS exposure ($F_{2,44}$ = 25.38, p < 0.0001), and a significant CS × drug interaction ($F_{2,44}$ = 3.66, p < 0.05). There were no other main effects or interactions ($F$'s < 1.6, p's > 0.05). Fisher's PLSD indicated that rats in the FW condition exhibited significantly more freezing during the retrieval test than rats in all other groups (p's < 0.05). BW-VEH rats exhibited significantly higher levels of freezing than BW-NBQX rats (p < 0.005); BW-NBQX rats did not significantly differ from the NoCS(Test) groups.

Given that freezing to the BW CS in the VEH-treated animals was maximal in the first half of the test, a separate factorial ANOVA was performed on the average percentage of freezing during trials 1–6 (Figure 1B). For this period, there was a main effect of CS treatment ($F_{2,44}$ = 18.61, p < 0.0001) as well as a CS treatment × drug interaction ($F_{2,44}$ = 3.81, p < 0.05); there was no main effect of drug across all groups ($F$ < 3.00, p > 0.05). Fisher's PLSD revealed significant differences in NBQX- and VEH-treated animals in the BW group (p < 0.0005). Additionally, rats in the NoCS(Test) groups differed from all others (p's < 0.0005) except those in the BW-NBQX group, indicating selective reduction of CS-elicited freezing in the BW-NBQX animals. It has been suggested that the BNST mediates sustained but not acute fear responses (*Davis, 2006*; *Davis et al., 2010*; *Walker et al., 2009*; *Walker and Davis, 2008*). However, it is notable that in this experiment BNST inactivation selectively attenuated conditioned freezing to the backward CS, which produced less sustained freezing during the retrieval test compared to the FW CS.

The different effects of BNST inactivation on freezing to FW and BW CSs may have been related to differences in memory strength between the two CSs (reflected in overall levels of freezing to each CS across the retrieval test). Indeed, to generate conditioned freezing to a backward CS we used more conditioning trials than typical in our laboratory. Of course, this may have resulted in moderate overtraining of the FW CS, which might have rendered the FW response less sensitive to disruption. To examine this possibility, we also examined the effects of BNST inactivation of FW and BW conditioning after five conditioning trials (*Figure 1—figure supplement 3*). The behavioral design for the five-trial experiment is shown in *Figure 1—figure supplement 3A*; cannula placements for this experiment are shown in *Figure 1—figure supplement 4*.

During the 5-trial conditioning procedure (*Figure 1—figure supplement 3B*), animals exhibited reliable increases in freezing across the session (repeated measures: $F_{5,115}$ = 12.154, p < 0.0001; there were no main effects of conditioning procedure or drug treatment and no interaction of these variables: $F$'s < 2.3, p's > 0.15). After infusions of NBQX or VEH, animals were tested to the CS in a familiar context that was distinct from the conditioning context. As shown in *Figure 1—figure supplement 3B*, BNST inactivation did not affect conditioned freezing to either the FW or BW CS. An ANOVA revealed a main effect of trial (repeated measures [including baseline freezing]: $F_{5,105}$ = 22.118, p < 0.0001), a main effect of conditioning procedure ($F_{1,21}$ = 15.930, p < 0.001), and a trial × conditioning procedure interaction (repeated measures: $F_{5,105}$ = 12.346, p < 0.0001). A separate ANOVA performed on trials 1–5 (excluding baseline; *Figure 1—figure supplement 3B*) revealed a significant main effect of conditioning procedure ($F_{1,21}$ = 21.585, p = 0.0001; there was no main effect of drug and no interaction: $F$'s < 0.15, p's > 0.7). Thus, the failure of BNST inactivation to impair conditioned freezing to the FW CS in the 12-trial procedure does not appear to be due to a ceiling effect; freezing to the FW CS in the 5-trial procedure was not maximal and remained insensitive to BNST inactivation. That said, the absence of an effect of BNST inactivation on freezing to the BW CS in this experiment may be due to a floor effect insofar as the 5-trial procedure did not yield conditional significant freezing to the CS (at least freezing that exceeded the pre-CS baseline). Together, these data indicate that the BNST is required for the expression of conditioned freezing to an excitatory BW, but not FW, CS.

One index that might reveal differences in the ability of the CS to predict the US is the US-evoked response itself. Footshock USs elicit an unconditioned response (UR) that includes vocalization, autonomic adjustments, and burst of locomotor activity (*Bali and Jaggi, 2015*; *Fanselow, 1994*). To better understand the mechanisms of forward and backward conditioning on behavior, we examined shock-evoked activity bursts (*Kunwar et al., 2015*; *Zelikowsky et al., 2018*) during the conditioning session in a separate cohort of animals (*Figure 1—figure supplement 5*). Fear conditioning resulted in robust freezing in both groups of animals (*Figure 1—figure supplement 5A*). An ANOVA revealed a significant main effect of trial (repeated measures: $F_{6,180}$ = 67.02, p < 0.0001) with no differences between levels of freezing in FW or BW animals (no other main effects and no interactions: $F$'s < 1.80, p's > 0.15). In contrast, shock-induced activity differed in FW and BW animals (*Figure 1—*

*figure supplement 5C*). All animals exhibited a reliable decrease in shock-induced activity across the conditioning session (repeated measures: $F_{5,150} = 9.59$, $p < 0.0001$) and the rate of decline in activity was similar in the two groups (no trial × group interaction: $F < 1.70$, $p > 0.09$), but the overall level of shock-induced activity was significantly lower in the FW animals (main effect of group: $F_{1,30} = 9.59$, $p < 0.05$; significant two-tailed unpaired $t$-test for mean levels: $t_{30} = -2.08$, $p < 0.05$). For comparison, two-tailed unpaired $t$-test of mean levels of activity during the 5 min baseline revealed no significant difference between FW and BW rats (*Figure 1—figure supplement 5B*; $t < 0.30$; $p > 0.75$). Hence, USs that were not signaled by a forward CS evoked greater activity bursts than those that were, which may suggest greater regulation of the shock-induced activity when the US is preceded by a temporally predictive cue.

## Inactivation of the BNST disrupts freezing to a discrete CS associated with random USs

The differential effect of BNST inactivation on freezing to a FW or BW CS suggests that the BNST regulates defensive behavior to stimuli that poorly signal US onset. However, the BW CS does not completely lack temporal information about US onset; the BW conditioning procedure may have resulted in forward trace conditioning to the CS, which always preceded the next US by 60 s (after trial 1). To address this possibility, separate rats were implanted with cannulas in the BNST and submitted to one of three conditioning procedures (*Figure 2*). In the first group, animals were conditioned to a backward CS (identical to 12-trial experiment above; 'BW'). In the second group, animals received CS presentations followed by randomized onset of the US ('RANDOM'). In the third control group, animals were conditioned using the RANDOM procedure but in the absence of any CS ['NoCS(Cond)"]. A summary of the behavioral design is shown in *Figure 2A* [the GABA$_A$ receptor agonist muscimol ('MUS') was used to reversibly inactivate the BNST in this and subsequent experiments; see Materials and methods]. Cannula placements for these animals are shown in *Figure 2— figure supplement 1*.

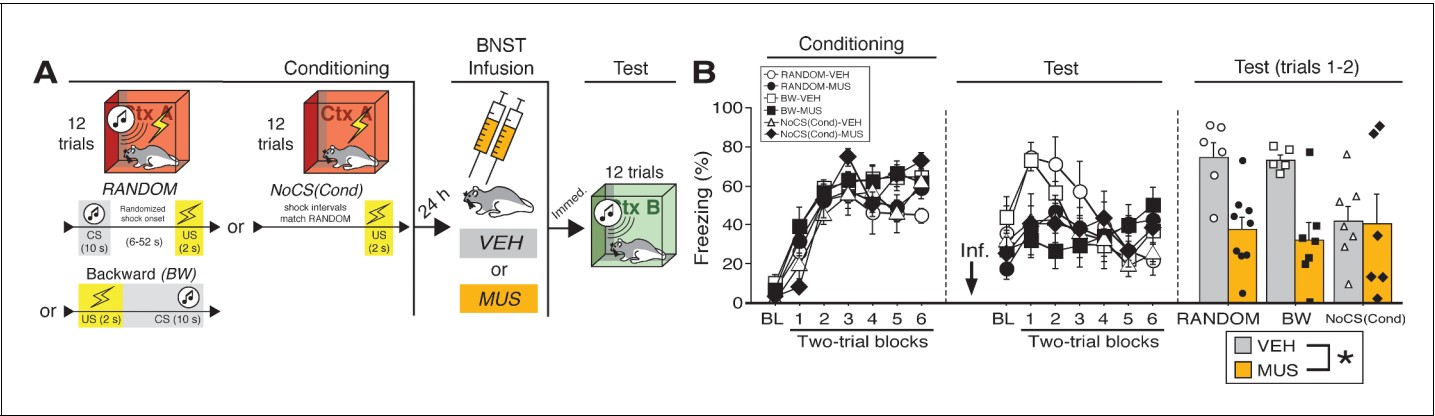

**Figure 2.** BNST inactivation attenuates freezing to a discrete CS paired with random onset of shock. (**A**) Behavioral schematic. (**B**) Freezing behavior during conditioning and retrieval testing. For conditioning, the left panel depicts mean percentage freezing during the 5 min baseline (BL) and across each conditioning block (each block is comprised of two trials; trials consist of freezing during the 10 s CS followed by the 58 s interval for the BW animals, whereas blocks for the other groups include equivalent time periods which include the CS [if present] and inter-CS intervals [excluding the duration of the shock]). For retrieval testing, the center panel shows mean percentage freezing at the 5 min baseline (BL) and across each test block (each 140 s block is comprised of two trials; trials consist of freezing during the 10 s CS followed by the 60 s interval). The right panel shows mean percentage freezing across the first two test trials (after BL; corresponding to 140 s of behavior). All data are represented as means ± s.e.m [RANDOM-VEH (n = 6); RANDOM-MUS (n = 9); BW-VEH (n = 5); BW-MUS (n = 7); NoCS(Cond)-VEH (n = 7); NoCS(Cond)-VEH (n = 6)]; * = p < 0.05.
DOI: https://doi.org/10.7554/eLife.46525.009

The following source data and figure supplement are available for figure 2:

**Source data 1.**
DOI: https://doi.org/10.7554/eLife.46525.011
**Figure supplement 1.** Bilateral cannula placements for BNST microinfusions.
DOI: https://doi.org/10.7554/eLife.46525.010

The percentage of freezing during fear conditioning for the RANDOM, BW, and NoCS(Cond) procedures is shown in *Figure 2B*. For conditioning, an ANOVA revealed a main effect of trial (repeated measures: $F_{6,204}$ = 85.209, p < 0.0001), insofar as all groups reliably increased their freezing across the conditioning trials. There was no main effect of conditioning procedure ($F$ < 3.10, p > 0.05) and no main effect or interactions with regards to future drug or vehicle assignments ($F$'s < 1.00, $p$'s > 0.30).

As shown in *Figure 2B*, inactivation of the BNST produced deficits in freezing during retrieval testing in the BW and RANDOM groups, but did not affect freezing in the NoCS(Cond) control. This was confirmed in an ANOVA that revealed a significant drug × trial interaction (repeated measures: $F_{6,204}$ = 7.91, p < 0.0001). A significant main effect of trial (repeated measures: $F_{6,204}$ = 7.80, p < 0.0001) indicated significant changes in freezing across the entire test amongst all groups. No other significant main effects or interactions were detected for this test ($F$'s < 1.80, $p$'s > 0.05). Similar to the aforementioned experiments, freezing was maximal in the early test trials, so a separate ANOVA was performed on mean responding during the first two trials of testing (*Figure 2B*). These analyses revealed a significant main effect of drug ($F_{1,34}$ = 12.82, p < 0.005), indicating a disruption of freezing in drug-treated animals; no other significant main effects or interactions were detected for these trials ($F$'s < 3.00, $p$'s > 0.07). Similar results were obtained if the ANOVA was restricted to the first six trials of the test (main effect of drug: $F_{1,34}$ = 6.217, p < 0.05; no other significant main effects or interactions, $F$'s < 1.90, $p$'s > 0.16). Excluding NoCS(Cond) animals, a main effect of drug treatment was also identified for the first two-trial block of the test ($F_{1,23}$ = 28.115, p < 0.0001). Additionally, we ran an ANOVA solely on baseline freezing. MUS-treated animals exhibited a trending, but nonsignificant ($F$ < 4.1, p > 0.05), decrease in freezing during the baseline [no other main effect or interaction was detected: ($F$'s < 1.00, $p$'s > 0.30)]. While baseline freezing may impact CS-responding, subtracting baseline responding from CS-elicited freezing (of the first two trials for RANDOM and BW animals) did not eliminate a main effect a drug-treatment ($F_{1,23}$ = 5.877, p < 0.05). Thus, these data replicate the effects of BNST inactivation on BW CS-elicited freezing as seen in our first experiment and extend these results to show that the BNST mediates freezing to CSs that signal uncertain latency of shock onset.

## Temporary inactivation of the BNST does not eliminate fear to a forward CS that is paired with a US of variable intensity

To determine whether the BNST is involved in other conditioning procedures imbued with outcome (but not temporal) uncertainty, we examined whether freezing to a FW CS that is paired with a US of variable intensity is also BNST-dependent. In this case, rats received forward fear conditioning with either a fixed ('FIXED') or variable ('VARIABLE') US intensity (*Figure 3*). A schematic of the behavioral design is shown in *Figure 3A*; cannula placements are illustrated in *Figure 3—figure supplement 1*. During conditioning (*Figure 3B*), ANOVA revealed a main effect of trial (repeated measures: $F_{6,168}$ = 68.15, p < 0.0001), with no main effect of drug or conditioning procedure, and no interactions ($F$'s < 1.50, $p$'s > 0.20).

Twenty-four hours after conditioning and immediately before a test to the CS in a novel context (*Figure 3B*), the animals were infused with either muscimol ('MUS') to reversibly inactivate the BNST or saline ('VEH') as a control. As shown in *Figure 3B*, BNST inactivation did not eliminate freezing to a FW CS paired with either a fixed or variable US. During retrieval testing, there was a main effect of trial (repeated measures: $F_{6,156}$ = 24.31, p < 0.0001), however no other main effects or interactions were detected ($F$'s < 2.70, $p$'s > 0.15). Although average baseline freezing was highest in VARIABLE-VEH animals, an ANOVA of baseline freezing did not reveal any main effects or interactions ($F$'s < 3.60, $p$'s > 0.05). To equate for differences in pre-CS freezing in the experimental groups, baseline freezing was subtracted from CS-elicited freezing (*Figure 3C*). This did not reveal an effect of BNST inactivation on freezing after baseline; there were no significant main effects or interactions in the ANOVA ($F$'s < 0.15, $p$'s > 0.70). In other words, the freezing in the presence of the CS was only increased relative to the baseline and was not masked by it. Although BNST inactivation trended towards reducing generalized contextual freezing in this experiment, it did not eliminate CS-elicited freezing to the temporally predictable cue, regardless of the variable magnitude of the US.

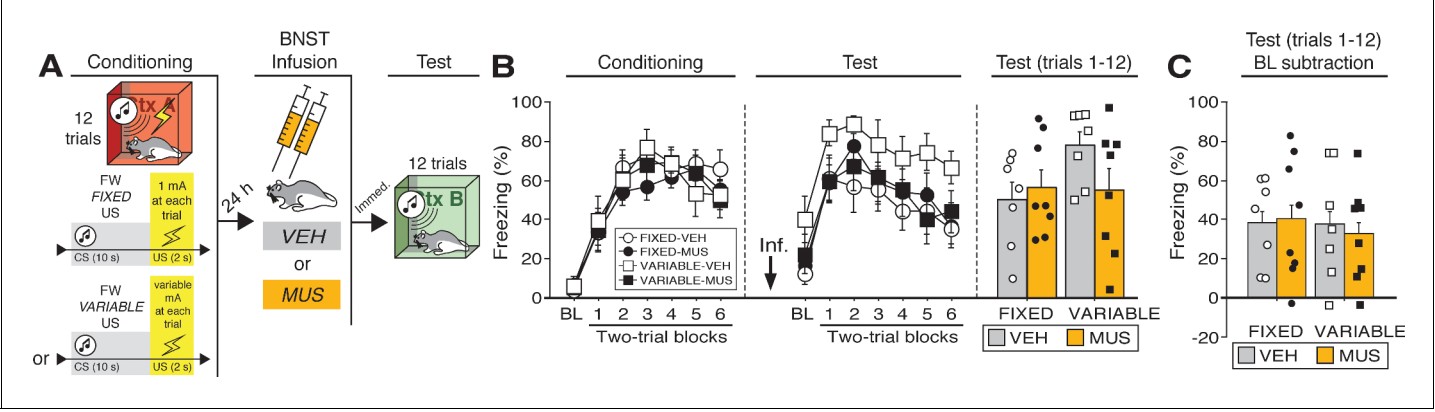

**Figure 3.** Temporary inactivation of the BNST does not prevent conditioned fear expression to a forward CS paired with a US of fixed or variable intensity. (A) Behavioral schematic. (B) Freezing behavior during conditioning and retrieval testing. For conditioning, the left panel depicts mean percentage freezing during the 5 min baseline (BL) and across each conditioning block (each 136 s block is comprised of two trials; trials consist of freezing during the 10 s CS followed by the 58 s post-shock interval). For retrieval testing, the center panel shows mean percentage freezing at the 5 min baseline (BL) and across each test block (each 140 s block is comprised of two trials; trials consist of freezing during the 10 s CS followed by the 60 s interval). The right panel shows mean percentage freezing across all test trials (after BL; corresponding to 840 s of behavior). (C) Freezing percentages across all twelve trials, with BL levels of freezing subtracted from these values. All data are represented as means ± s.e.m [FIXED-VEH (*n* = 7); FIXED-MUS (*n* = 8); VARIABLE-VEH (*n* = 7); VARIABLE-MUS (*n* = 8)].

DOI: https://doi.org/10.7554/eLife.46525.012

The following source data and figure supplement are available for figure 3:

**Source data 1.**

DOI: https://doi.org/10.7554/eLife.46525.014

**Figure supplement 1.** Bilateral cannula placements for BNST microinfusions.

DOI: https://doi.org/10.7554/eLife.46525.013

## Backward CSs selectively increase fos expression in the ventral BNST

To further examine the role of the BNST in the expression of conditioned fear, we quantified Fos expression in multiple subregions of the BNST following the presentation of either a FW or BW CS during a shock-free retrieval test (*Figure 4*). The behavioral design is summarized in *Figure 4A*. Four experimental groups were compared: rats conditioned and tested to a forward CS ('FW'), rats conditioned and tested to a backward CS ('BW'), rats conditioned to a forward or backward CS but not receiving a CS at test ['NoCS(Test)"], and animals that were conditioned but not tested ('NoTest'). Conditioning (*Figure 4B*) was similar to previous experiments (main effect of trial; repeated measures: $F_{6,258} = 77.346$, p < 0.0001; no other main effects or interactions: *F*'s < 1.30, *p*'s > 0.30). As shown in *Figure 4B*, the groups differed in their levels of conditioned freezing during the retrieval test. ANOVA revealed a main effect of trial (repeated measures: $F_{6,216} = 15.54$, p < 0.0001), conditioning procedure ($F_{2,36} = 11.42$, p < 0.0001), and a procedure × trial interaction (repeated measures: $F_{6,216} = 3.65$, p<0.0001) (includes baseline). Post-hoc analyses revealed that FW (p < 0.0001) and BW (p < 0.005) rats exhibited significantly higher levels of freezing behavior than NoCS(Test) rats. Similarly, an ANOVA of average freezing across the test trials (without baseline) revealed a main effect of group ($F_{2,36} = 13.76$, p < 0.0001). Post-hoc comparisons revealed significant differences between FW vs. BW (p < 0.05) and FW vs. NoCS(Test) rats (p < 0.0001), as well as BW vs. NoCS(Test) animals (p < 0.005).

Ninety minutes after the retrieval test, the animals were sacrificed for Fos immunohistochemistry. Fos-positive nuclei were counted in three BNST subregions (*Figure 5A*): 'ovBNST' (Fos counts targeting the oval nucleus of the BNST), 'dBNST' (counts in an area containing the dorsal region of the anteromedial BNST), and 'vBNST' [Fos counts in a region containing the BNST's anterolateral, fusiform, and anteromedial (ventral) nuclei; refer to *Swanson (2003)*. As shown in *Figure 5B*, the average number of Fos-positive nuclei for each of these regions differed in rats undergoing FW or BW conditioning in the ovBNST and vBNST, but not the dBNST. In the vBNST, an ANOVA revealed a main effect of group ($F_{3,43} = 11.41$, p < 0.0001). Fisher's PLSD revealed that BW rats exhibited

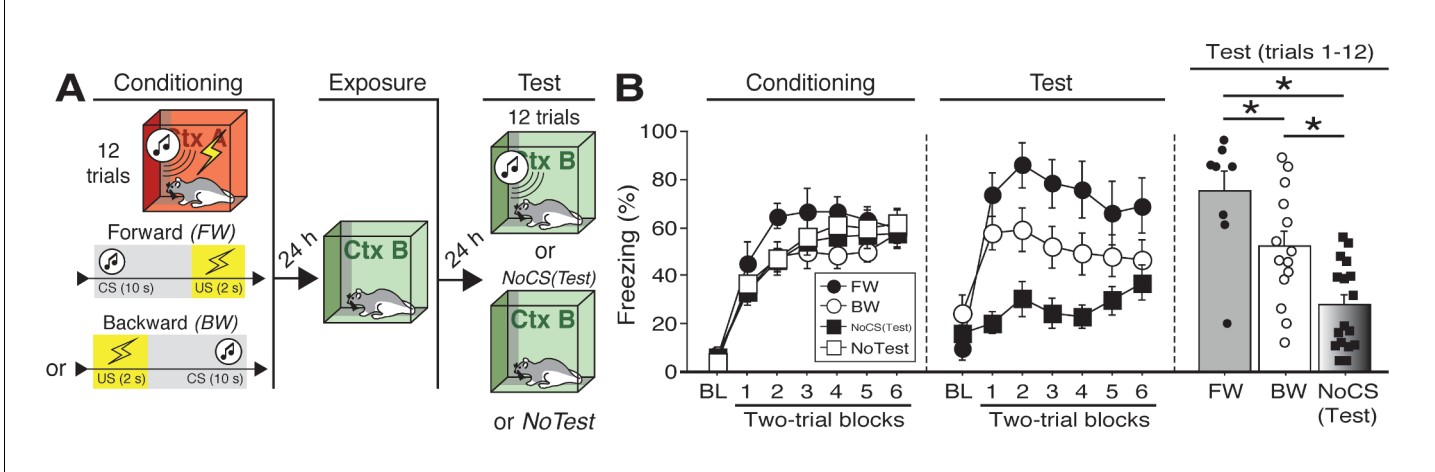

**Figure 4.** CS-evoked freezing in rats utilized for Fos analyses. (**A**) Behavioral schematic. (**B**) Freezing during conditioning and retrieval testing. For conditioning, the left panel depicts mean percentage freezing during the 5 min baseline (BL) and across each conditioning block (each 136 s block is comprised of two trials; conditioning trials consist of freezing during the 10 s CS followed by the 58 s interstimulus interval). For retrieval testing, the center panel shows mean percentage freezing at the 5 min baseline (BL) and across each test block (each 140 s block is comprised of two trials; trials consist of freezing during the 10 s CS followed by the 60 s interval). The right panel shows mean percentage freezing after BL (corresponding to 840 s of behavior). Animals were sacrificed for Fos analyses 90 min after trial 1. All data are represented as means ± s.e.m [FW ($n = 8$); BW ($n = 14$); NoCS (Test) ($n = 17$); NoTest ($n = 8$)]; * = $p < 0.05$.
DOI: https://doi.org/10.7554/eLife.46525.015
The following source data is available for figure 4:

**Source data 1.**
DOI: https://doi.org/10.7554/eLife.46525.016

significantly higher levels of Fos compared to FW ($p < 0.0005$), NoCS(Test) ($p < 0.001$), and NoTest ($p < 0.0001$) rats. Additionally, NoCS(Test) rats exhibited significantly greater Fos expression than NoTest rats ($p < 0.05$). In the ovBNST, an ANOVA revealed a main effect of group ($F_{3,43} = 3.26$, $p < 0.05$). Post-hoc analyses revealed that NoTest rats exhibited significantly higher Fos levels

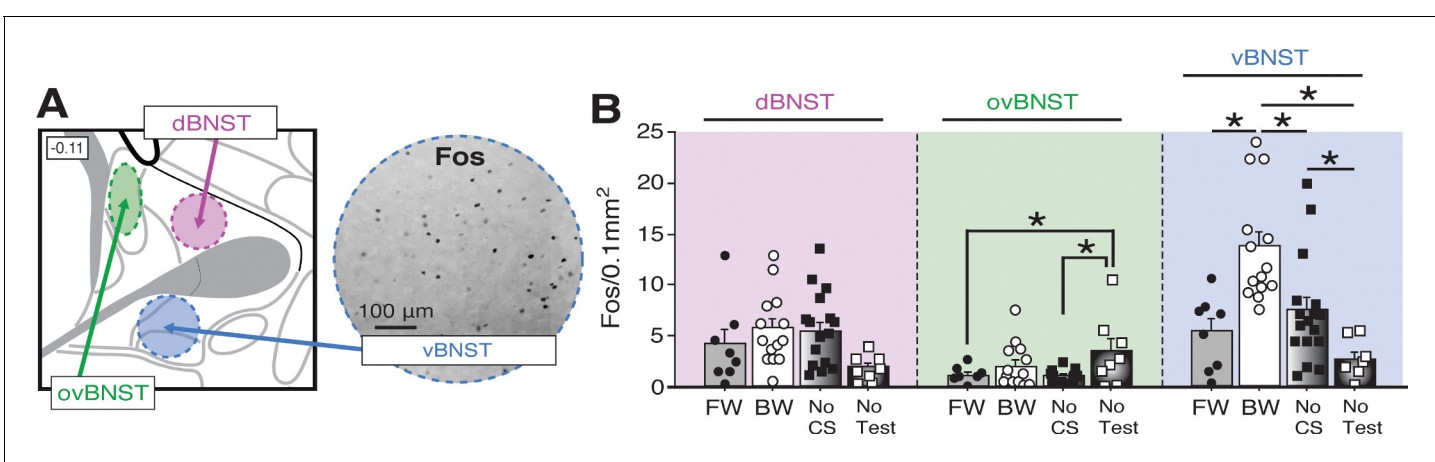

**Figure 5.** Fos expression in the BNST following exposure to a temporally predictable or uncertain CS. (**A**) Schematic depicting regions counted within the BNST (left panel). The right panel shows example of Fos expression in the ventral BNST (vBNST). (**B**) Mean Fos-positive cells per 0.1 mm$^2$ for each of the quantified regions. All data are represented as means ± s.e.m [FW ($n = 8$); BW ($n = 14$); NoCS(Test) ($n = 17$); NoTest ($n = 8$)]; * = $p < 0.05$.
DOI: https://doi.org/10.7554/eLife.46525.017
The following source data is available for figure 5:

**Source data 1.**
DOI: https://doi.org/10.7554/eLife.46525.018

compared to FW (p < 0.05) and No-CS (p < 0.01) rats. In dBNST, there was no effect of conditioning procedure on Fos expression (factorial ANOVA: $F$ < 2.60, p > 0.07). Together, these data indicate that presentation of a BW CS increases Fos expression in the vBNST. Moreover, exposure to the temporally predictive FW CS (which elicited the highest levels of fear) was associated with low levels of Fos in the BNST.

### Backward CSs selectively increase fos expression in mPFC afferents of the BNST

The BNST receives input from many areas involved in the regulation of fear (*Fox and Shackman, 2019*; *Goode and Maren, 2017*; *Lebow and Chen, 2016*). Therefore, also we used a functional tracing procedure to quantify Fos expression in neurons targeting the BNST (*Figure 6*). Specifically, rats were injected with a retrograde tracer (CTb-488) into the BNST (prior to conditioning) so that we could quantify Fos expression in BNST-projecting neurons in its major limbic and cortical afferents. These were the same subjects used for the BNST Fos analyses (see prior section). So, in addition to Fos in the BNST, we quantified fear-elicited Fos in the infralimbic (IL) and prelimbic (PL) regions of the medial prefrontal cortex (mPFC), the basolateral amygdala (BLA), and the ventral hippocampus (HPC). The behavioral data for these animals are shown in *Figure 4*.

A representative image of CTb infusion into the BNST is shown in *Figure 6A* and an illustration of the largest and smallest CTb spread of injection included in the analyses is shown in *Figure 6B*. Approximate microinjection sites for CTb for all animals are shown in *Figure 6—figure supplement 1*. Representative Fos and CTb co-labeling is shown in *Figure 6C*. As shown in *Figure 6D*, there were reliable differences in CTb labeling among the BNST afferents we quantified. An ANOVA revealed a significant main effect of region ($F_{3,116}$ = 42.34, p < 0.0001). Post-hoc comparisons indicated that the number of CTb-positive cells in the HPC were significantly higher than that in all other regions (p < 0.0001, per comparison); IL and BLA exhibited significantly greater CTb counts compared to PL (p < 0.0001, per comparison). CTb counts in each region did not differ by behavioral

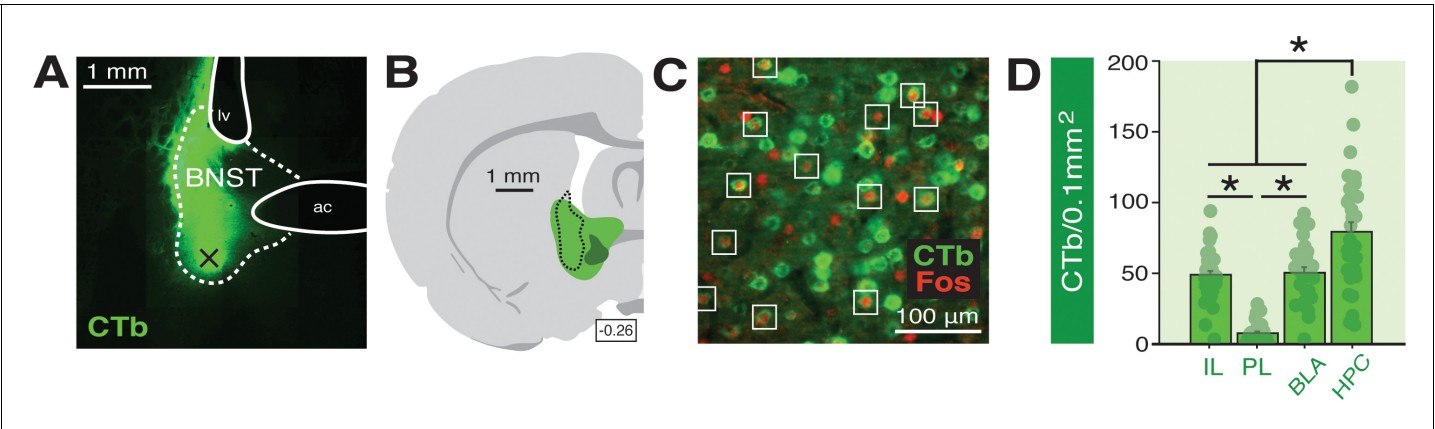

**Figure 6.** Functional tracing in afferents targeting the BNST. (**A**) Coronal section (10×) showing representative fluorescence of a CTb infusion (green) into the BNST (dotted outline; 'ac'=anterior commissure, 'lv'=lateral ventricle). Black 'X' denotes approximate lowest point of the infusion (as an example of how injection sites are documented in *Figure 6—figure supplement 1*). (**B**) Coronal schematic (−0.26 mm from bregma) showing the approximate largest (green) and smallest (dark green) areas of CTb spread in the BNST for animals included in the analyses; the black dotted outline represents the extent of spread of CTb in the BNST in the image shown in panel A. (**C**) Example CTb-positive (green) and Fos-positive (red; nuclei) cells in a coronal section (40 μm) of the IL; open white squares denote double-labeled cells. (**D**) Mean number of BNST-targeting CTb-positive cells (per 0.1 mm²) for each of the quantified regions (shows FW, BW, and NoTest animals corresponding to *Figure 7*). All data are represented as means ± s.e.m (for each region, *n* = 30); * = *p* < 0.05.

DOI: https://doi.org/10.7554/eLife.46525.019

The following source data and figure supplement are available for figure 6:

**Source data 1.**
DOI: https://doi.org/10.7554/eLife.46525.021
**Figure supplement 1.** CTb injection sites in BNST.
DOI: https://doi.org/10.7554/eLife.46525.020

condition ($F < 0.2$, $p > 0.85$) and were collapsed for *Figure 6D*. These data indicate extensive connectivity of the PFC, BLA, and HPC with the BNST.

As shown in *Figure 7A*, the number of Fos-positive nuclei in the IL, PL, HPC, and BLA differed among the behavioral groups. Factorial ANOVA of Fos counts in IL revealed a main effect of group ($F_{2,27} = 8.55$, $p < 0.005$). Post-hoc comparisons for IL revealed significant differences between BW vs. NoTest rats ($p < 0.0005$) and FW vs. NoTest rats ($p < 0.05$). In PL, a main effect of group was also identified ($F_{2,27} = 6.79$, $p < 0.005$). Post-hoc analyses indicated that BW rats exhibited significantly more Fos in PL as compared to NoTest animals ($p<0.001$). For BLA, a main effect of group was detected (factorial ANOVA: $F_{2,27} = 8.10$, $p < 0.005$). Post-hoc analyses revealed that FW and BW rats (which did not significantly differ) exhibited significantly more Fos in BLA as compared to No Test rats ($p < 0.01$, FW vs. NoTest; $p < 0.001$, BW vs. NoTest). In HPC, an ANOVA revealed a significant main effect of group ($F_{2,27} = 4.28$, $p < 0.05$). Post-hoc comparisons indicated that FW and BW rats (again, which did not significantly differ) had significantly more Fos expression in HPC as compared to NoTest animals ($p < 0.05$ for FW vs. NoTest and BW vs. NoTest). These data indicate that

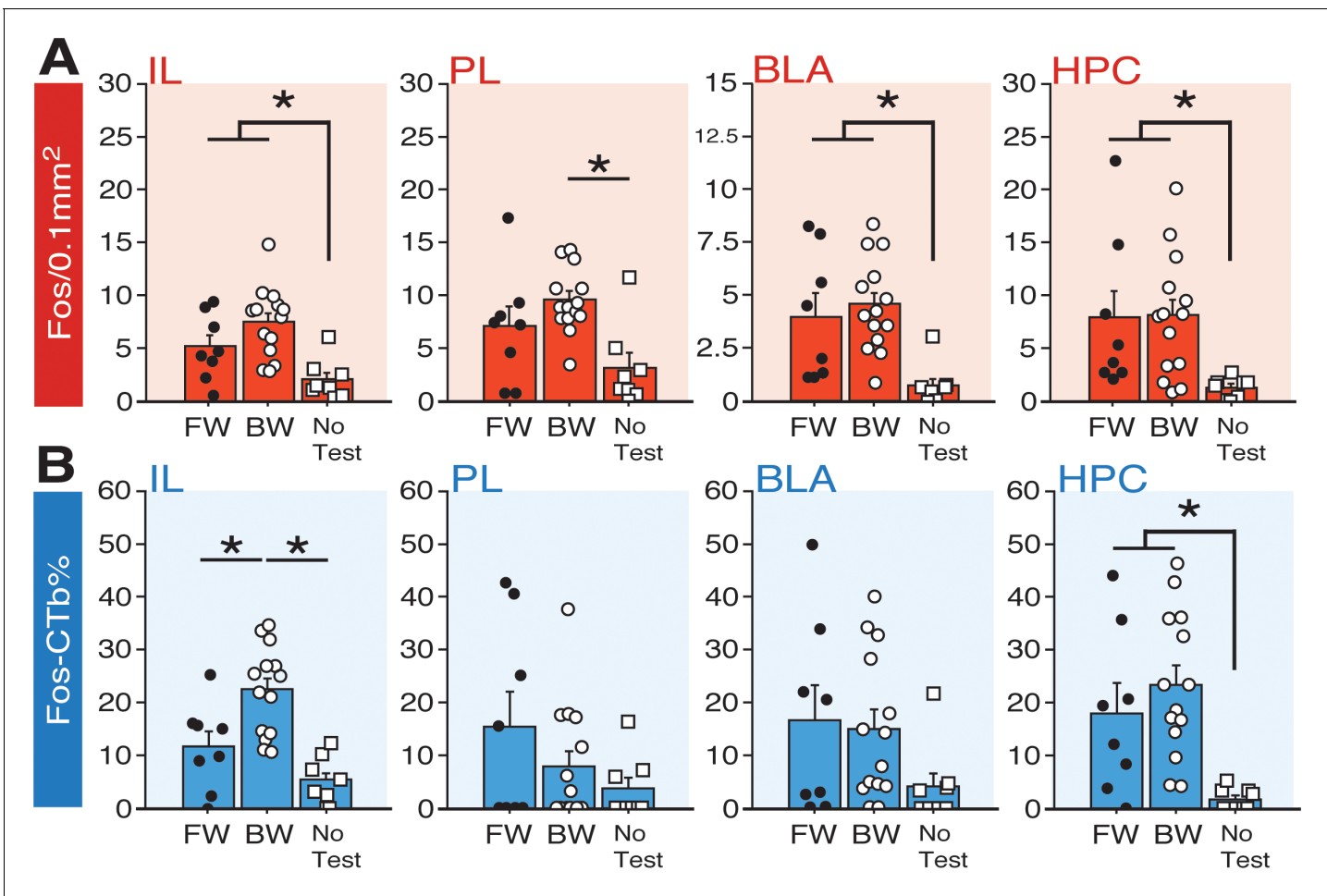

**Figure 7.** Fos expression in BNST-targeting cells of the prefrontal cortex, amygdala, and hippocampus following exposure to a forward or backward CS. (A) Mean number of Fos-positive cells (per 0.1 mm$^2$) for each of the quantified regions. (B) Mean percentage (per 0.1 mm$^2$) of Fos-positive and CTb-positive cells divided by the total number of CTb-positive cells for each region. All data are represented as means ± s.e.m [FW ($n = 8$); BW ($n = 14$); NoTest ($n = 8$)]; * = $p < 0.05$.
DOI: https://doi.org/10.7554/eLife.46525.022
The following source data is available for figure 7:

**Source data 1.**
DOI: https://doi.org/10.7554/eLife.46525.023

conditioned freezing to forward or backward CSs increased the number of Fos-positive neurons in the PFC, BLA, and HPC.

To quantify the fraction of BNST-projecting neurons within the PFC, BLA, and HPC that were activated by a FW or BW CS, we calculated a ratio of Fos-positive to CTb-positive nuclei in each afferent region ('Fos-CTb%'; *Figure 7B*). Interestingly, within the IL, animals in the BW group exhibited the highest fraction of double-labeled cells. An ANOVA revealed a significant main effect of group ($F_{2,27}$ = 14.22, p < 0.0001). Post-hoc comparisons revealed significant differences between BW and FW rats (p < 0.005) as well as BW and NoTest animals (p < 0.0001). In contrast, there were no differences between FW and BW rats in any of the other regions quantified. In PL, there was not a significant main effect of group ($F$ < 1.90, p > 0.17) and no group effects were observed in the BLA ($F$ < 2.0, p > 0.15). In the HPC, rats in the FW and BW groups exhibited greater numbers of double-labeled cells than those in the NoTest group ($F_{2,27}$ = 7.82, p < 0.01). Additionally, we examined whether freezing significantly correlated with any of the aforementioned Fos measures; gross measures of freezing percentages (trials 1–12) for FW and BW rats did not significantly correlate with Fos levels across the multiple regions (data not shown in figures), suggesting that the presence or absence of the fear CS was more predictive of activity in these circuits than freezing performance per se. Nonetheless, these data indicate that conditioned freezing (in both FW and BW rats) is associated with increased activity in BNST-targeting cells of HPC, and that BW CSs selectively increase Fos in BNST-projecting neurons in IL. Hence, IL projections to the BNST may regulate the effects of BW CSs on BNST Fos expression and freezing.

## Discussion

In the present study, we demonstrate that the BNST is critical for processing unpredictable threats, specifically discrete auditory CSs that are poor predictors of when aversive footshocks will occur. Reversible inactivation of the BNST disrupted CS-elicited freezing to a backward CS, but did not prevent freezing to a CS that reliably signaled the imminent onset of shock, even when that CS signaled an unpredictable intensity of the US. Moreover, pharmacological blockade of the BNST attenuated fear to both a backward CS and a CS that was trained with random US onset. These data suggest that uncertain timing of shock is a critical factor in recruiting the BNST to conditioned fear. Moreover, BNST inactivation did not affect freezing to a forward CS that was paired with a US that varied in intensity, suggesting that contributions of the BNST to conditioned fear may depend on the type of uncertainty imbued by the CS. Interestingly, freezing to the backward CS was less sustained than that to the forward CS, suggesting that neither long-duration stimuli nor sustained freezing responses are required to recruit the BNST. Finally, although both forward and backward CSs increased Fos expression in the hippocampus, BNST, amygdala, and mPFC, only backward CSs selectively increased Fos expression in both the ventral BNST and BNST-projecting neurons in the infralimbic cortex. Collectively, these data reveal that the BNST and its mPFC afferents may mediate defensive responding to threats that signal uncertain or remote onset of shock.

Unpredictable threats come in many forms [including uncertain timing, magnitude or type, or probability of threat: (*Bennett et al., 2018*; *Daldrup et al., 2015*; *Davies and Craske, 2015*; *McHugh et al., 2015*; *McNally et al., 2011*; *Schroijen et al., 2016*; *Seidenbecher et al., 2016*); these subtypes of uncertainty may have dissociable and cumulative contributions to anxiety. In particular, uncertainty about the timing of the onset of aversive events has been linked to measures of anxiety in humans and other animals (*Amadi et al., 2017*; *Bennett et al., 2018*; *Shankman et al., 2011*). Interestingly, unpredictable threats (of varying types and degrees) are associated with increases or changes in regional cerebral blood flow and functional connectivity in the primate BNST (*Alvarez et al., 2015*; *Alvarez et al., 2011*; *Brinkmann et al., 2018*; *Brinkmann et al., 2017a*; *Brinkmann et al., 2017b*; *Buff et al., 2017*; *Choi et al., 2014*; *Choi et al., 2012*; *Coaster et al., 2011*; *Fox et al., 2018*; *Grupe et al., 2013*; *Herrmann et al., 2016*; *Kalin et al., 2008*; *Kalin et al., 2005*; *Klumpers et al., 2017*; *Klumpers et al., 2015*; *McMenamin et al., 2014*; *Meyer et al., 2019*; *Mobbs et al., 2010*; *Münsterkötter et al., 2015*; *Naaz et al., 2019*; *Pedersen et al., 2018*; *Pedersen et al., 2017*; *Schlund et al., 2013*; *Sladky et al., 2018*; *Somerville et al., 2010*; *Torrisi et al., 2018*). In turn, BNST lesions or circuit inactivations reduce defensive responding to contextual cues that poorly signal the imminence of the aversive US (*Asok et al., 2018*; *Davis and Walker, 2014*; *Goode et al., 2015*; *Hammack et al., 2015*; *Luyten et al., 2011*; *Poulos et al.,*

*2010*; *Sullivan et al., 2004*; *Waddell et al., 2006*; *Zimmerman and Maren, 2011*; but, see *Luyck et al., 2018a*). Consistent with the possibility that the BNST mediates conditioned fear to temporally uncertain threats, we observed significant and selective decreases in the expression of defensive freezing when we inactivated the BNST in the presence of BW CSs or CSs trained with randomized USs. Conversely, cues that predicted imminent shock were not affected by BNST inactivation. Additionally, we observed significant differences in shock reactivity during forward and backward conditioning, insofar as rats exhibited more robust shock-elicited activity bursts than during forward conditioning. These data may suggest that conditioning in the presence of predictive cues may reduce the magnitude of the unconditioned response, in a way that is not present for the backward CS.

A critical finding in the present study is that the BNST mediates conditioned freezing to short-duration discrete CSs. This contrasts with prior work suggesting that the BNST only mediates freezing behavior to long-duration cues or contexts (*Hammack et al., 2015*; *Waddell et al., 2006*). However, recent data in other paradigms suggest a role for the BNST in the processing of responses to relatively brief threat stimuli (*Brinkmann et al., 2018*; *Choi et al., 2014*; *Haufler et al., 2013*; *Kinnison et al., 2012*; *Kiyokawa et al., 2015*; *Klumpers et al., 2017*; *Luyck et al., 2018a*; *Luyck et al., 2018b*; *Naaz et al., 2019*). Together, these data suggest that stimulus duration is not necessarily a determinant of BNST involvement. Similarly, it does not appear that response duration is the primary determinant of the BNST in threat reactions. In the present work, conditioned freezing to the backward CS was generally shorter-lived than that to the forward CS, but was more sensitive to BNST inactivation than freezing to the forward CS. One might conclude from this observation that the BNST mediates weak (but not strong) freezing responses. However, we and others have shown that BNST lesions or inactivation reduce the expression of freezing even when those levels are relatively high (*Goode et al., 2015*; *Hammack et al., 2015*). Hence, the BNST mediates fear to cues that signal uncertain or remote threat, independent of the duration of the antecedent stimuli or behavioral responses associated with the threat (*Goode and Maren, 2017*).

Interestingly, it has been suggested that fear to a backward CS relies on associations between the CS and the conditioning context (*Chang et al., 2003*), which more strongly predicts the US. That is, extinction of fear to the conditioning context reduces fear to a BW, but not FW, CS. Thus, it is possible that the BNST mediates fear expression to a backward CS by retrieving a memory of the context-US association. Indeed, considerable work indicates that BNST is involved in the expression of contextual fear (*Davis and Walker, 2014*; *Goode et al., 2015*; *Hammack et al., 2015*; *Luyten et al., 2011*; *Sullivan et al., 2004*; *Waddell et al., 2006*; *Zimmerman and Maren, 2011*). Increases in immediate early gene expression, metabolic activity, and changes in electrophysiological firing patterns have been observed in the BNST in response to direct exposure to conditioned contextual cues (*Campeau et al., 1997*; *Daldrup et al., 2016*; *Jennings et al., 2013*; *Lemos et al., 2010*; *Luyten et al., 2012*; *Marcinkiewcz et al., 2016*). If the backward CS functions similarly to a conditioned context, then one might expect Fos activity to be elevated in the BNST of animals exposed to the backward CS. Indeed, we observed significantly more Fos expression in the BNST in response to the backward compared to the forward CS. The backward CS-elicited Fos expression was specific to vBNST, insofar as backward and forward rats did not differ in overall levels of Fos in dBNST or ovBNST. Similarly, others have shown elevated Fos expression, in regions in or near vBNST after stress or conditioned fear-related behavior (*Besnard et al., 2019*; *Sterrenburg et al., 2012*; *Verma et al., 2018*; *Radley and Johnson, 2018*; *Radley and Sawchenko, 2011*).

Of course, another possibility is that the backward conditioning procedure yielded forward trace conditioning after the first trial (*Burman et al., 2014*; *Marchand et al., 2004*; *Raybuck and Lattal, 2014*; *Tipps et al., 2014*). Although there has been considerable work on the neural mechanisms of trace conditioning (*Raybuck and Lattal, 2014*), a role for the BNST in this form of learning has not yet been established. Because trace conditioning also degrades the temporal imminence of the US following the CS, it might be expected to require the BNST. Consistent with this possibility, we found that inactivation of the BNST reduced fear in the presence of a cue that was conditioned with randomized onset of shock after the CS. In either event, the BNST appears to be recruited by stimuli, whether cues or contexts, that signal remote or temporally unpredictable onset of shock.

The amygdala, PFC, and hippocampus have strong connections with the BNST (*Canteras and Swanson, 1992*); *deCampo and Fudge, 2013*; *Dong et al., 2001*; *Glangetas et al., 2017*; *Johnson et al., 2019*; *McDonald et al., 1999*; *Reichard et al., 2017*; *Reynolds and Zahm, 2005*;

*Torrisi et al., 2015*; *Vertes, 2004*; *Weller and Smith, 1982*; *Wood et al., 2019*). In line with previous work, the expression of freezing (whether elicited by a forward or, in our case, backward CS) increased the number of Fos-positive neurons in these regions (*Herry et al., 2008*; *Jin and Maren, 2015*; *Knapska and Maren, 2009*; *Lemos et al., 2010*; *Wang et al., 2016*). In addition, we found that forward and backward CSs increased Fos expression in BNST-targeting cells of the HPC (with similar, but trending, effects within the BLA). HPC and BLA projections to the BNST are thought to regulate hypothalamic-pituitary-adrenal (HPA) axis activity (*Crestani et al., 2013*; *Forray and Gysling, 2004*; *Zhu et al., 2001*), and these afferents may be engaged in high-fear states to mobilize corticosterone release, for example (*Kim et al., 2013*). However, unlike the HPC and BLA, we observed a greater number of Fos-positive cells in the IL in response to a backward CS relative to forward CS. This suggests a unique role for BNST-projecting neurons of the IL [which appear to be glutamatergic: (*Crowley et al., 2016*; *Glangetas et al., 2017*)] in processing excitatory backwards CSs. There is growing evidence for a regulatory role for prefrontal connections to the BNST in stress and anxiety-like behavior (*Fox et al., 2010*; *Glangetas et al., 2017*; *Glangetas et al., 2013*; *Johnson et al., 2019*; *Kinnison et al., 2012*; *Motzkin et al., 2015*; *Naaz et al., 2019*; *Radley et al., 2009*), the present data adds to this work by suggesting that the IL may be involved in generating freezing and/or stress responding to the backward CS. This would stand in contrast to considerable data indicating that the IL is involved in the inhibition of freezing behavior after extinction, for example (*Milad and Quirk, 2002*; *Quirk and Mueller, 2008*). However, the fact that the backward CS elicits less fear (or more extinguishable fear) could suggest this circuit is active, in part, to gate or regulate fear expression. Indeed, there is evidence for a role of the infralimbic cortex in regulating discrimination of conditioned excitors and inhibitors (*Sangha et al., 2014*). An alternative view is that the IL is involved in processing contextual information conveyed by the hippocampus (*Corcoran and Quirk, 2007*; *Marek et al., 2018*; *Zelikowsky et al., 2013*) during threat processing. To the extent that that backward CSs elicit fear via CS-(context-US) associations (*Chang et al., 2003*), it is possible that the IL is involved in retrieving CS-context associations.

In conclusion, we demonstrate a novel role for the BNST and its circuits in processing discrete cues, including excitatory backward CSs and randomized CSs. Specifically, BNST inactivation impaired conditioned freezing to a backward, but not forward, auditory CS that differed only in their temporal relationship to a footshock US. This reveals that neither stimulus modality nor duration (or response duration) are the critical parameters driving BNST involvement in defensive behavior. We further show that backward CSs increase the number of Fos-positive neurons in the vBNST and in BNST-projecting neurons in the infralimbic cortex. This suggests a novel role for mPFC projections to the BNST in processing unpredictable threats, possibly of the aversive contexts to which backward CSs are associated. These findings may help in our understanding of the broader contributions of the BNST to motivated behaviors that depend on uncertainty, such as during BNST-dependent triggers of fear and drug relapse (*Aston-Jones and Harris, 2004*; *Avery et al., 2016*; *Goode et al., 2018*; *Goode and Maren, 2019*; *Harris and Winder, 2018*; *Miles et al., 2018*; *Silberman and Winder, 2013*; *Stamatakis et al., 2014*). In total, these experiments provide new insight into the antecedents for BNST-dependent defensive behavior and highlight behavioral procedures to explore these processes.

## Materials and methods

### Subjects

All experiments used adult (200–240 g upon arrival; *n* = 285, before exclusions) male Long-Evans rats (Envigo; Indianapolis, IN). Rats were housed in a climate-controlled vivarium and kept on a fixed light/dark cycle (lights on starting at 7:00 AM and off at 9:00 PM; experiments took place during the light phase of the cycle). Rats were individually housed in clear plastic cages (with bedding consisting of wood shavings; changed weekly) on a rotating cage rack. Group assignments for behavioral testing were randomized for cage position on the racks. Animals had access to standard rodent chow and water *ad libitum*. Animals were handled by the experimenter(s) (~30 sec/day) for five consecutive days prior to the start of any surgeries or behavior. All procedures were in accordance with the US National Institutes of Health (NIH) Guide for the Care and Use of Laboratory Animals and were approved by the Texas A&M University Institutional Animal Care and Use Committee.

## Apparatuses

All behavioral testing occurred within distinct rooms in the laboratory. Each behavioral room housed eight identical rodent conditioning chambers (30 cm × 24 cm × 21 cm; MED Associates, Inc). Each chamber was housed in a larger, external sound-attenuating cabinet. Rear walls, ceilings, and the front doors of the testing chambers were made of Plexiglas, while their sidewalls were made of aluminum. Grid floors of the chambers were comprised of nineteen stainless steel bars (4 mm in diameter), and spaced 1.5 cm apart (center to center). The grid floors were attached to an electric shock source and a solid-state grid scrambler for delivery of the US (MED Associates, Inc). A speaker attached to each chamber was used to deliver the auditory CS. As needed for each context, the chambers were equipped with 15 W house lights, and small fans were embedded in the cabinets (providing background noise of ~70 dB). An aluminum pan was inserted beneath the grid floor to collect animal waste. A small camera was attached to the top of the cabinet for video monitoring of behavior.

Measurements of freezing were performed using an automated and unbiased system (*Maren, 1998*). Specifically, each behavioral testing chamber rested on a load-cell platform that was sensitive to cage displacement due to the animal's movements. During behavioral testing, load-cell activity values (ranging from −10 to +10 V) were collected and digitized at 5 Hz using Threshold Activity Software (MED Associates, Inc). Offline conversions of the load-cell activity values were performed to generate absolute values ranging from 0 to 100; lower values indicate minimal cage displacement, which coincided with freezing behaviors in the chambers. Accordingly, freezing bouts were defined as absolute values of ≤10 for 1 s or more. The percentage of freezing behavior during the pre-CS baseline, the CS, and the inter-trial intervals was computed for each behavioral session. Shock reactivity was analyzed by directly reporting the absolute values generated by the Threshold Activity Software (i.e., larger values indicated more movement in the cage).

Unique contexts (A and B) were used for conditioning retrieval testing. Chamber assignments were unique to each context and group assignments were counterbalanced across test chambers when possible. For each experiment, context A was assigned to one of the behavioral rooms, and B the other. For context A, the test chamber was wiped down with an acetic acid solution (3%) and a small amount was poured in the pans beneath the grid floors. The cage lights were turned on, while the chamber fans were turned off. The cabinet doors were closed. The behavioral room was illuminated with dim red light. Animals were transported to and from the chambers using white plastic transport boxes. For context B, an ammonium hydroxide solution (1%) was used to wipe down and scent the chambers, black Plexiglas panels were placed over the grid floors, the cage lights were turned off, the chamber fans were turned on, and the cabinet doors remained open. The behavioral room was illuminated with white light (red room lights were turned off). Rats were transported to and from the context using black plastic transport boxes that included a layer of clean bedding.

## Surgeries

For animals receiving intracranial microinfusions into the BNST, rats were transported to the surgical suite and deeply anesthetized using isoflurane (5% for induction, 1–2% for maintenance) (*Acca et al., 2017*; *Goode et al., 2015*; *Nagaya et al., 2015*; *Zimmerman and Maren, 2011*). Rats were then secured in a stereotaxic frame (Kopf Instruments). Hair on top of the rodent's head was shaved, povidone-iodine was applied, and a small incision was made in the scalp to expose the skull. Bregma and lambda were aligned on a horizontal plane, and small holes were drilled in the skull for the placement of anchoring screws and bilateral stainless-steel guide cannulas (26 gauge, 8 mm from the bottom of their plastic pedestals; Small Parts). The cannulas were inserted into the BNST at the following coordinates: −0.15 mm posterior to bregma, ±2.65 mm lateral to the midline, and −5.85 mm dorsal to dura (guide cannulas were angled at 10° with their needles directed at the midline). Dental cement was used to secure the cannulas to the screws and stainless steel obturators (33 gauge, 9 mm; Small Parts) were inserted into the guide cannulas. Animals were given at least one week to recover prior to the onset of behavioral training.

For rats injected with cholera toxin subunit B (CTb) conjugated with Alexa Fluor-488 (CTb-488; ThermoFisher Scientific) in the BNST, the surgical procedures were similar to those described above. A single small hole was drilled into the skull to allow for the insertion of a glass micropipette. Injection tips were backfilled with mineral oil and secured in the injector; CTb-488 was then drawn up

into the injector immediately before use. Rats received unilateral CTb-488 infusions into either the left or right hemisphere (group assignments were randomized for sites of CTb-488 infusion): −0.15 mm posterior to bregma, ±2.65 mm lateral to the midline, and −6.50 mm dorsal to dura (the pipette was angled at 10° with the tip directed at the midline). CTb-488 (5.0 mg/µl; total volume of 0.25 µl) was microinfused into the brain using a Nanoject II auto-nanoliter injector (Drummond Scientific Co) secured to the arms of the stereotaxic frame. For the infusion process, 50 nl (25 nl/s) of CTb-488 was infused once per min for 5 min to achieve 0.25 µl of the total infusion (the injection needle was left in the brain for five additional minutes to allow for diffusion of CTb-488). Following the infusion procedures, the incision was closed with sutures and the animals were returned to their homecages where they recovered for ~10 days.

## Intracranial infusions

Prior to behavioral testing, the animals were acclimated to the process of intracranial infusions. On two occasions, the animals were transported from the vivarium to the infusion room in 5-gallon buckets containing a layer of bedding; the experimenter(s) removed the obturators and replaced them with clean ones. On the day of infusions, animals (in predetermined squads of four to eight rats; representing all drug/behavioral groups as possible) were transported to the laboratory, obturators were removed, and injectors were inserted into the guides. Stainless steel injectors (33 gauge, Small Parts; 9 mm extending 1 mm beyond the end of the guide cannula) were connected to polyethylene tubing (PE-20; Braintree Scientific); the other end of the tubing was connected to gastight 10 µl syringes (Hamilton, Co). The syringes were mounted to an infusion pump (KD Scientific, Inc). For the experiments shown in *Figure 1* and *Figure 1—figure supplement 3*, the α-amino-3-hydroxy-5-methyl-4-isoxazolepropionic acid (AMPA) receptor antagonist 2,3-dihydroxy-6-nitro-7-sulfamoyl-benzo[f]quinoxaline-2,3-dione (NBQX) was used to reversibly inactivate the BNST (*Adami et al., 2017*; *Davis and Walker, 2014*; *Goode et al., 2015*; *Zimmerman and Maren, 2011*). NBQX disodium salt hydrate (Sigma Life Sciences) was dissolved in saline to a concentration of 10.0 µg/µl ('NBQX'); physiological saline served as the vehicle ('VEH' for all experiments). For the experiments shown in *Figures 2* and *3*, the γ-aminobutyric acid (GABA)$_A$ receptor agonist muscimol was used to reversibly inactivate the BNST (*Bangasser et al., 2005*; *Breitfeld et al., 2015*; *Buffalari and See, 2011*; *Fendt et al., 2003*; *Goode et al., 2015*; *Markham et al., 2009*; *Pina et al., 2015*; *Sajdyk et al., 2008*; *Xu et al., 2012*). Muscimol (Sigma-Aldrich) was dissolved in physiological saline to a concentration of 0.1 µg/µl ('MUS' for *Figures 3* and *5*); physiological saline served as the vehicle ('VEH'). Although both drugs have been used to inactivate the BNST (see above), we used NBQX and muscimol in separate experiments to observe whether one was more effective in attenuating FW- or BW-elicited behavior; both drugs were effective in reducing BW CS-elicited freezing. For the representative image showing drug spread in the BNST (*Figure 1—figure supplement 1*), muscimol TMR-X conjugate (Thermo Fisher Scientific) was dissolved in physiological saline to a concentration of 0.1 µg/µl and used for infusions. For all of the aforementioned experiments, drug or vehicle was drawn into the injectors (immediately prior to the infusions) and a total volume of 0.275 µl/hemisphere of drug or vehicle was infused at a rate of 0.275 µl/min; injectors were left in the cannulas for 1 min following the infusions to allow for diffusion. Once the injectors were removed, clean obturators were inserted into the guides.

## Behavioral procedures

Overviews of each behavioral experiment are provided in the figures. The conditioned stimulus (CS) for all experiments was an auditory tone (80 dB, 2 kHz, 10 s), which was paired with the unconditioned stimulus (US, footshock; 1.0 mA, 2 s). For the variable shock intensity experiment (*Figure 4*), US intensity varied as described.

### FW/BW/NoCS(Test) BNST inactivation (twelve training trials)

In a 3 × 2 design, animals (n = 64, prior to exclusions) were randomly assigned to receive a CS at test [either a forward ('FW')- or backward ('BW')-trained CS] or no CS at test [animals were trained to a BW CS; 'No-CS'], and NBQX ('NBQX') or vehicle infusions ('VEH'). Of these rats, seven were excluded due to off-target cannulas, three additional rats were excluded due to illness, and four more were excluded due to a technical error during infusions that resulted in no drug or vehicle to

be infused (fourteen total exclusions). This resulted in the following (final) group numbers (shown in data/figures): FW-VEH ($n$ = 5); FW-NBQX ($n$ = 4); BW-VEH ($n$ = 13); BW-NBQX ($n$ = 12); No-CS-VEH ($n$ = 8); No-CS-NBQX ($n$ = 8). For conditioning, animals (in squads of eight rats) were transported from the vivarium to context A; FW and BW procedures were run in alternating squads of animals. Drug and vehicle assignments were counterbalanced for position in the chambers. For FW conditioning, were placed in the context for 5 min before the delivery of twelve CS-US trials (CS offset immediately preceded US onset; 70 s intertrial interval). After the final conditioning trial, the animals remained in the chamber for 1 min before being returned to their homecages (the entire conditioning session consisted of 19 min total; for both FW and BW conditioning). For BW conditioning, all aspects of conditioning were identical to the FW training except the order of the CS and US was reversed. That is, after a 5 min baseline period, the animals received 12 US-CS trials (the onset of the CS occurred immediately after the offset of the US; 70 s intertrial intervals). After conditioning, rats were returned to their homecages.

Twenty-four hours after conditioning, animals (in squads of four) were infused with NBQX or VEH into the BNST immediately before being placed in context B. After 5 min of acclimation to the context, FW and BW animals (intermixed in each squad) received twelve presentations of the CS in the absence of the US (70 s intertrial interval). The rats remained in the chambers for 1 min following the final CS (19 min session, in total). For rats not receiving the CS at test, the animals remained in the test context for 19 min without presentation of the CS or US. We alternated running squads that received the CS and those that did not after the infusions. Following the test, animals were returned to their homecages.

## FW/BW BNST inactivation (five training trials)

In a 2 × 2 design, animals ($n$ = 28, prior to exclusions) were randomly assigned to receive forward ('FW') or backward ('BW') conditioning using five training trials, and NBQX ('NBQX') or vehicle infusions ('VEH'). Of these rats, three were excluded due to off-target cannulas. This resulted in the following (final) group numbers (shown in data/figures): FW-NBQX ($n$ = 6); FW-VEH ($n$ = 6); BW-NBQX ($n$ = 6); BW-VEH ($n$ = 7). At the start of behavior, rats (in squads of seven) were transported from the vivarium and placed in context A. We alternated squads that received forward or backward conditioning. For forward conditioning, rats were given 3 min of acclimation to the context prior to the onset of five CS-US trials. Rats remained in the chamber for 1 min following the final trial (530 s total conditioning session). For backward conditioning, rats were placed in context a for a 3 min baseline (530 s session total), after which they received five US-CS trials. Rats were returned to their homecages after conditioning. Twenty-four hours after conditioning, the animals were placed in context B in the absence of the CS or US for 530 s to acclimate them to the test context. After this acclimation session, rats were again returned to their homecages.

Forty-eight hours after conditioning, the animals received NBQX or VEH infusions (identical to above) immediately prior to retrieval testing. FW and BW animals were intermixed in each squad. Immediately after the infusions, animals were placed in context B for a retrieval test to assess freezing to the CS. The test (530 s in total) consisted of a 3 min baseline period followed by five CS-alone presentations (70 s intertrial interval). Rats were returned to their homecages following the test session.

## FW/BW intra-shock reactivity

Rats ($n$ = 32, no exclusions) were randomly assigned to receive forward ('FW') or backward ('BW') conditioning. No rats were excluded from this experiment (no infusions occurred; $n$ = 16, per group; only data from the conditioning session are shown. Animals (in squads of eight) were transported to context A for either FW or BW conditioning. Parameters for FW and BW conditioning were identical to the procedures for the previous 12-trial experiments. We alternated FW and BW squads. Rats were returned to their homecages following training.

## RANDOM/BW/NoCS(Cond) BNST inactivation

In a 3 × 2 design, rats ($n$ = 70, prior to exclusions) were randomly assigned to receive backward conditioning ('BW'; identical to the 12-trial BW conditioning described above), randomized CS/US trials in which the duration of the intervals between CS offset and US onset vary on each trial

('RANDOM'), or US-only contextual conditioning [shock intervals matched the RANDOM group; 'NoCS(Cond)"]. Of these groups, animals were randomly assigned to receive muscimol ('MUS') or vehicle ('VEH') infusions immediately prior to retrieval. For this experiment, thirty animals were excluded due to off-target cannulas. This resulted in the following (final) group numbers (shown in data/figures): BW-VEH ($n$ = 5); BW-MUS ($n$ = 7); RANDOM-VEH ($n$ = 6); RANDOM-MUS ($n$ = 9); NoCS(Cond)-VEH ($n$ = 7); NoCS(Cond)-MUS ($n$ = 6).

For conditioning, animals (in squads of five to eight rats; counterbalanced for the type of conditioning) were transported to context A and underwent BW, RANDOM, or NoCS(Cond) training. Drug and vehicle assignments were counterbalanced for the conditioning procedure and chamber placements for each squad. For BW animals, the conditioning parameters matched those used for the 12-trial experiment described above. For RANDOM animals, and after the 5 min baseline, CS presentations matched the 60 s interstimulus intervals used for the 12-trial FW paradigms (noted above); US onset occurred at the following intervals after offset of each CS at each trial: 26 s, 52 s, 10 s, 40 s, 16 s, 52 s, 14 s, 50 s, 18 s, 36 s, 48 s, 6 s (the entire session last 19 min). For NoCS(Cond) animals, rats experienced the same protocol as the RANDOM group, but no auditory CS occurred. Rats were returned to their homecages following conditioning.

Twenty-four hours later, animals (in squads of four to six) were transported to the laboratory to receive pre-retrieval infusions of MUS or VEH (counterbalanced in each squad). BW, RANDOM, and NoCS(Cond) animals were intermixed in each squad. Immediately after the intracranial infusions, and after a 5 min baseline in context B, each squad of rats received twelve presentations of the CS in the absence of the US. Each CS presentation was separated by 70 s intertrial intervals, with the entire test lasting 19 min. Rats were returned to their homecages following the test session.

## FIXED/VARIABLE BNST inactivation

In a 2 × 2 design, rats ($n$ = 31, prior to exclusions) were randomly assigned to receive forward conditioning with consistent ('FIXED') or variable magnitudes of the US ('VARIABLE'); muscimol ('MUS') or vehicle ('VEH') infusions were made prior to retrieval testing. For FIXED animals, US intensity was 1 mA at each trial. For VARIABLE animals, shock intensity (in mA) varied across trials in this order (mean = 1 mA): 0.5, 1.8, 0.4, 1.6, 1.4, 0.3, 0.5, 1.8, 0.4, 1.6, 1.4, 0.3. In this experiment, one rat was excluded due to off-target cannulas. This resulted in the following (final) group numbers (shown in data/figures): FIXED-VEH ($n$ = 7); FIXED-MUS ($n$ = 8); VARIABLE-VEH ($n$ = 7); VARIABLE-MUS ($n$ = 8).

For conditioning, animals (in squads of seven to eight) were transported to context A (squads alternated between FIXED and VARIABLE paradigms). The rats were given 5 min to acclimate to the context before the onset of 12 CS-US pairings (70 s intertrial intervals). Rats remained in the chambers for 1 min following the final CS, with the entire conditioning session lasting 19 min. Rats were returned to their homecages following conditioning.

Twenty-four hours later, rats (in squads of seven to eight) received intracranial infusions of MUS or VEH into the BNST immediately before being placed in context B for retrieval testing. Rats were given a 5 min baseline period before the onset of twelve CS-alone presentations (70 s intertrial intervals). Rats remained in the chambers for 1 min following the final CS, with the entire test session lasting 19 min. Rats were returned to their homecages following the test.

## FW/BW Fos-CTb

Animals ($n$ = 60, before exclusions) were randomly assigned to receive a forward ('FW')- or backward ('BW')-trained CS at testing, or no CS retrieval at test ['NoCS(Test)"]. Note that the NoCS(Test) group consists of animals that were trained to either a FW or BW CS. Additionally, a group of BW-trained animals remained in their homecages ('NoTest') during the final test and were sacrificed alongside the other groups. Twelve rats were excluded for either excessive or off-target infusion of CTb-488 outside the borders of the BNST or an absence of CTb-488 labeling in the ventral BNST. One additional rat was excluded due to a technical issue that resulted in the loss of tissue at the level of the prefrontal cortex. This resulted in the following (final) group numbers (shown in data/figures): FW ($n$ = 8); BW ($n$ = 14); NoCS(Test) [$n$ = 17 (BW-trained: $n$ = 9; FW-trained: $n$ = 8)]; NoTest ($n$ = 8). For behavioral training, rats (in squads of six to eight; with all groups intermixed) were transported to the laboratory and placed in context B to acclimate for 5 min (no CS or US) and then

returned to their homecages. Later that day, the animals were transported to context A where they received twelve trials of either FW or BW fear conditioning, which was identical to the other afore-mentioned procedures. We alternated FW and BW conditioning squads. After conditioning, the rats were returned to their homecages. Twenty-four hours after conditioning, all rats were exposed to context B for 20 min in the absence of the CS or US to acclimate them to the test context.

Forty-eight hours after conditioning, FW, BW, and NoCS(Test) rats were transported to context B to receive a retrieval test to the CS or were merely placed in the context [NoCS(Test) condition]. Squads alternated between the CS and no CS test. For rats undergoing CS retrieval, FW and BW animals (intermixed in each squad) received CS trials as described previously. Rats were perfused 90 min following the first CS of the test, in groups of three to four. For NoCS(Test) rats (with FW- and BW-trained animals intermixed), animals were exposed to context B in the absence of the CS or US. NoCS(Test) rats were perfused 95 min after being placed in the test context, in groups of three to four. NoTest rats were perfused alongside groups of FW, BW, and NoCS(Test) rats. Rats were returned to their homecages after testing and prior to the perfusions.

## Histological procedures

At the conclusion of behavioral testing, cannula-implanted animals were overdosed with sodium pentobarbital (Fatal Plus; 100 mg/ml, 0.5 ml, i.p.). Transcardial perfusions were then performed using chilled physiological saline followed by 10% formalin. Brains were extracted and stored in 10% formalin for 24 hr at 4° C; brains were transferred to a 30% sucrose-formalin solution for three or more days (at 4° C) before sectioning. Brains were flash frozen with crushed dry ice and coronal sections (40 μm) containing the BNST were collected using a cryostat (Leica Microsystems) at −20° C. The tissue was wet-mounted to gelatin-subbed microscope slides and stained with 0.25% thionin prior to adding glass coverslips secured with mounting medium (Permount, Sigma). To further examine the spread of drug, a subset of animals was infused (identical to the aforementioned infusion parameters) with fluorescent muscimol (1.0 μg/μl; EverFluor TMR-X conjugate, Setareh Biotech) before being sacrificed (these animals were not perfused) and having brains dissected (40 μm; brains were stored in 30% sucrose solution at 4° C until sectioning). These tissues were wet-mounted to slides and aqueous mounting medium (Fluoromount; Sigma-Aldrich) was used to secure glass coverslips.

For CTb-injected animals, post-behavior perfusions mirrored the aforementioned procedures. For sectioning, coronal sections (which included regions of the prefrontal cortex, BNST, basolateral amygdala, and ventral hippocampus) were collected into well plates containing phosphate-buffered saline (1× PBS) and stored in the dark at 4° C until immunohistochemistry could be performed. For localization of the CTb-injection site, separate sections at the level of the BNST were wet-mounted and coverslipped using Fluoromount mounting medium.

## Immunohistochemistry

Immunohistochemistry for Fos was performed on free-floating brain tissue similar to prior reports (*Jin and Maren, 2015*; *Marek et al., 2018*; *Orsini et al., 2011*; *Wang et al., 2016*). For sections containing the BNST, Fos was stained using the following procedures [all steps were performed at room temperature (~20° C) on a shaker, unless stated otherwise; rinses were brief (~20 s)]. The tissue was first rinsed in 1× tris buffered saline (TBS; 7.4 pH), and then incubated in 0.3% $H_2O_2$ (in TBS) for 15 min, followed by rinses (×3) in TBS. Slices were transferred to primary antibody [rabbit anti-c-fos, 1:10,000 in 1× TBS containing Tween 20 (TBST); Millipore] and incubated overnight. Sections were then rinsed (×3) in TBS before incubating in secondary antibody for 1 hr (biotinylated goat anti-rabbit, 1:1000 in TBST; Jackson Immunoresearch). Sections were rinsed (×3) again in TBS. The slices were transferred to wells containing avidin biotin complex (ABC, 1:1000 in TBST; Vector Labs) for 45 min. The tissues were again rinsed (×3) in TBS. Tissue was transferred to wells containing 3,3′ diaminobenzidine [(DAB) 5% stock, 1:200], nickel ammonium sulfate (5% stock, 1:10), and 30% $H_2O_2$ (1:2,000) in TBS for 10 min to generate purple/black nuclear chromophore products. After another rinse (×3) in TBS, the tissue was subsequently wet-mounted to slides and secured with coverslips using Permount mounting medium.

For sections containing the prefrontal cortex, amygdala, and hippocampus (Fos-CTb experiment), Fos was stained using the following procedures [unless stated otherwise, each step occurred at

room temperature (~20° C) on a shaker (and away from excess light)]. First, the tissue was rinsed (10 min; ×2) in 1× TBS, followed by a 10 min wash in 1× TBST. The tissue was incubated in 10% normal donkey serum (NDS; in TBST) for 1 hr. The slices were then rinsed (5 min; ×2) in TBST. Sections were transferred to primary antibody [goat anti-c-fos, 1:2000 in 3% NDS (in TBST); Santa Cruz Biotechnology] and incubated on a rotator in the dark for 72 hr at 4° C. The tissue was rinsed (10 min; ×3) in TBST before incubating in secondary antibody [biotinylated donkey anti-goat, 1:200 in 3% NDS (in TBST); Santa Cruz Biotechnology] for 2 hr. Slices were then rinsed (10 min; ×3) in TBST. Sections were transferred to wells containing streptavidin (Alexa-Fluor 594-conjugate, 1:500 in 3% NDS (in TBST); Thermo Fisher Scientific] for 1 hr. Tissue was rinsed (10 min; ×3) in TBS before being wet-mounted to slides and secured with coverslips using Fluoromount mounting medium.

## Image analyses

All imaging and counting procedures (for all regions) were performed with the experimenter(s) blind to the group assignments of the animals. For thionin-stained coronal tissue, photomicrographs of cannula in the BNST were generated (10× magnification) using a Leica microscope (MZFLIII) and Leica Firecam software. For animals infused with fluorescent muscimol into the BNST, infusion sites were imaged (10× magnification) using a Zeiss microscope and Axio Imager software (Zen Pro 2012). For CTb-injected animals, BNST images were generated (10× magnification) using the same Zeiss microscope and software.

For Fos analyses in BNST, brightfield images of BNST [in BNST regions ranging from approximately −0.00 to −0.50 mm posterior to bregma of the skull, and from both left and right hemispheres (randomized for site of CTb-injection)] were generated (10× magnification) using a Zeiss microscope and Axio Imager software (Zen Pro 2012). ImageJ software (*Schneider et al., 2012*) was used to count cells. Counts were confined to the following areas of interest: (1) 'ovBNST' [an area of 0.217 mm × 0.558 mm (oval); targeting the oval nucleus of the BNST], (2) 'dBNST' [0.372 mm$^2$ (circle); targeting the dorsal and medial subregions of the anterior BNST, including the anteromedial area of the BNST (dorsal to the anterior commissure)], and (3) 'vBNST' [0.434 mm$^2$ (circle); targeting the ventral regions (ventral to the anterior commissure) of the anterior BNST, which includes the ventral portion of the anteromedial area, the anterolateral area, and the fusiform nucleus of the BNST] (*Swanson, 2003*). For each of these regions, three to six images were quantified and averaged for each animal (Fos levels were standardized to 0.1 mm$^2$).

For Fos-CTb analyses in the prefrontal cortex, amygdala, and hippocampus, images (10× magnification) of Fos and CTb expression were generated using a Zeiss microscope and Axio Imager software (Zen Pro 2012). Cell counts were analyzed using ImageJ software. Images were generated and analyzed only for the hemisphere injected with CTb. Counts were confined to the following areas of interest: (1) 'IL' [an area of 0.805 mm × 0.217 mm (rectangle); targeting deep and superficial layers of the infralimbic cortex (approximately +2.0 to +3.2 mm anterior to bregma of the skull)], (2) 'PL' [1.115 mm × 0.217 mm (rectangle); targeting deep and superficial layers of the pre-limbic cortex (approximately +2.0 to +3.2 mm anterior to bregma of the skull)], (3) 'BLA' [0.434 mm$^2$ (circle); targeting the basolateral amygdala (approximately −1.8 to −3.5 mm posterior to bregma of the skull)], and (4) 'HPC' [0.496 mm × 0.217 mm (rectangle); targeting the ventral subiculum but may include some CA1 cells of the ventral hippocampus (approximately within −4.7 to −7.0 mm posterior to bregma of the skull)] (*Swanson, 2003*). For each of these regions, three to six images were quantified and averaged for each animal (all counts and percentages were normalized to 0.1 mm$^2$).

## Statistics

All data were submitted to repeated or factorial analysis of variance (ANOVA) or two-tailed *t*-tests as described for each experiment. Fisher's protected least significant difference (PLSD) test was used for *post hoc* comparisons of group means following a significant omnibus *F* ratio in the ANOVA (α was set at 0.05). No statistical methods were used to predetermine group sizes (group sizes were selected based on prior work and what is common for the field). Data distributions were assumed to be normal, but these were not formally tested. Unless stated otherwise, all data are represented as means ± s.e.m.

## Acknowledgements

The authors thank Carolyn Evemy, Kaitlyn French, and Sohmee Kim for technical assistance. Supported by grants from the National Institutes of Health (R01MH065961 and R01MH117852 to SM and F31MH107113 to TDG), as well as a McKnight Foundation Memory and Cognitive Disorders Award and a Brain and Behavior Research Foundation NARSAD Distinguished Investigator Grant to SM.

## Additional information

### Funding

| Funder | Grant reference number | Author |
|---|---|---|
| National Institute of Mental Health | R01MH065961 | Stephen Maren |
| McKnight Endowment Fund for Neuroscience | Memory and Cognitive Disorders Award | Stephen Maren |
| Brain and Behavior Research Foundation | Distinguished Investigator Grant | Stephen Maren |
| National Institute of Mental Health | R01MH117852 | Stephen Maren |
| National Institute of Mental Health | F31MH107113 | Travis D Goode |

The funders had no role in study design, data collection and interpretation, or the decision to submit the work for publication.

### Author contributions

Travis D Goode, Conceptualization, Formal analysis, Funding acquisition, Investigation, Methodology, Writing—original draft, Writing—review and editing, Designed the research, Performed the research, Analyzed the data; Reed L Ressler, Olivia W Miles, Investigation, Performed the research, Analyzed the data, Assisted with preparing the manuscript; Gillian M Acca, Investigation, Methodology, Performed the research, Analyzed the data, Assisted with preparing the manuscript; Stephen Maren, Conceptualization, Resources, Data curation, Formal analysis, Supervision, Funding acquisition, Investigation, Methodology, Writing—original draft, Project administration, Writing—review and editing, designed the research, Analyzed the data

### Author ORCIDs

Travis D Goode http://orcid.org/0000-0003-1432-8894
Reed L Ressler https://orcid.org/0000-0003-0514-8269
Stephen Maren http://orcid.org/0000-0002-9342-7411

### Ethics

Animal experimentation: This study was performed in strict accordance with the recommendations in the Guide for the Care and Use of Laboratory Animals of the National Institutes of Health. All of the animals were handled according to approved institutional animal care and use committee (IACUC) protocols (#2015-005) of Texas A&M University. All surgery was performed under isoflurane anesthesia, and every effort was made to minimize suffering.

### Decision letter and Author response

Decision letter https://doi.org/10.7554/eLife.46525.026
Author response https://doi.org/10.7554/eLife.46525.027

## Additional files

### Supplementary files

• Transparent reporting form
DOI: https://doi.org/10.7554/eLife.46525.024

### Data availability

All data generated or analyses during this study are included in the manuscript and supporting files.

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
