## [Decision Letter]

[Editors’ note: a previous version of this study was rejected after peer review, but the authors submitted for reconsideration. The first decision letter after peer review is shown below.]

Thank you for submitting your work entitled "Bed nucleus of the stria terminalis mediates fear to ambiguous threat signals" for consideration by *eLife*. Your article has been reviewed by three peer reviewers, one of whom is a member of our Board of Reviewing Editors, and the evaluation has been overseen by a Senior Editor.

Our decision has been reached after consultation between the reviewers. Based on these discussions and the individual reviews below, we regret to inform you that your work will not be considered further for publication in *eLife*.

The primary problem, which is mentioned in each review but came out more strongly as an issue requiring further experimentation in the online discussion, is the lack of clear evidence that backward and forward conditioning differ primarily in the uncertainty of the prediction. Each reviewer felt this problem in the manuscript, though for each it is expressed a bit differently. In the end it was felt that addressing this, as well as other concerns about the strength of some of the results, would require more experiments. As *eLife* has a policy not to invite revisions if they require more than a couple months of experiments, the decision was made to reject. However, the reviewers all agreed that these were potentially very interesting experiments, so if the additional work were done, they were open to the authors coming back, essentially as a resubmission.

*Reviewer #1:*

In the current study, the authors present a series of nicely controlled and straightforward experiments comparing the effects of BNST inactivation on expression of backward and forward conditioned fear responses. The underlying idea was to elucidate the role of the BNST in responses to cues that were predictors of an aversive US versus those that were uncertain predictors. A backward conditioning design was used for the latter as a way to maintain some uncertainty while avoiding confounding changes in the duration of the cue or the strength of the US, its replicability, etc. This is a nice aspect of the design. The authors show clearly that inactivation impairs freezing after BW but not FW conditioning. The lack of effect is not due to differences in the strength of conditioning for FW, since FW was not affected with when lighter conditioning was used. Nor was it affected when a variable US was employed. Finally, the authors show that BW also induces Fos expression in parts of BNST and afferent prefrontal regions, not affected by FW. Overall this provides a very nice set of experiments differentiating the role of this circuit versus amygdalar circuits in these two forms of conditioning.

What I am less convinced of is that the involvement of the BNST in BW is due to some special role in uncertainty. To some extent this reflects my ignorance regarding BW and how it is conceptualized. In some other studies, procedures such as these are used to examine safety signaling for example. This is not to say I do not see the authors' point; it seems to me that this is something that might be given more explanation in the Introduction (which currently is all about BNST).

Perhaps related to this, I am a bit unclear on the significance of the data in Figure 3. This seems to be presented to substantiate that BW is making the subjects fearful but uncertain – is this correct? I wonder if showing this earlier might help support the paradigms. Or showing some direct relationship with the efficacy of BNST inactivation would make this relationship more clear. That is, if a failure to reduce the activity bursts to the US with conditioning reflects uncertainty about the shock and this is mediated by BNST, then the less this reduction, the more BNST should be required for the conditioned responding no? Likewise, it would be useful to know whether there were any relationships within groups between behavior and Fos expression in the various regions.

Reviewer #2:

In a series of experiments, Goode and colleagues examined the role of the bed nucleus of the stria terminalis (BNST) in a traditional fear conditioning arrangement (forward conditioning: tone → shock) and the inverse arrangement (backward conditioning: shock → tone). The logic behind this approach was that, historically, BNST manipulations have had little/no impact in traditional fear conditioning procedures. However, BNST is essential to fear driven by contexts or long duration stimuli – both of which introduce a measure of temporal uncertainty. Goode and colleagues speculated that backward fear conditioning may also introduce a degree of uncertainty, and therefore be sensitive to BNST manipulations. Consistent with their predictions, the authors demonstrate that BNST inactivation with NBQX selectively impaired 24 hour fear recall of the backward conditioned tone, but not the forward conditioned tone. This effect was not observed with fewer forward and backward conditioning trials nor did BNST inactivation impair fear recall to a forward conditioned cue associated with variable foot shock intensity. Using c-Fos, the authors demonstrate that the backward conditioned cue selectively activated the ventral subdivision of the BNST, particularly BNST cells receiving direct inputs from the infralimbic cortex.

The inactivation, tracing and c-Fos imaging methods are all appropriate and are the strength of the manuscript. The innovation comes from the use of backward conditioning. This is a behavioral procedure with a rich history in learning theory but has received little or no attention from neuroscientists. However, no effort is made to draw from this tradition. Instead, the authors immediately cast backwards conditioning as capturing 'threat uncertainty' and 'ambiguous threat' with no convincing explanation. Even more, there are Pavlovian fear conditioning procedures that directly manipulate ambiguity [Rescorla, (1968); McHugh et al., (2015); Erlich, Bush and Ledoux, (2012)]. Thus, there is a strong mismatch between the process the authors set out to investigate, and the processes captured by backward conditioning. The best general advice I can give is for the authors to search for and thoroughly read backward conditioning papers by Heth, Rescorla, Wagner, Ayres, LoLordo, Domjan and their contemporaries.

Essential revisions:

1) In experiment 1 the authors demonstrate the backward conditioning yields less fear, or more extinguishable fear, compared to a forward conditioned cue. Backward fear is also greater in rats given the cue during test, compared to rats given no cue. There is still a concern that rather than producing weak excitatory conditioning, freezing produced by the backward cue is non-associative. That is, equivalent levels of freezing would be observed to a cue unpaired with foot shock. Showing that backward conditioned rats expressed greater freezing in test, compared to unpaired rats, would definitively demonstrate that freezing was the result of associative learning.

2) ANOVA for Experiment 1 was adequately performed and the main effects & interactions reflect the patterns in the data. However, some issues with post-hoc testing crept up. The following must be a typo, as a group cannot differ from itself:

Results section: "Importantly, BW-VEH rats exhibited significantly higher levels of freezing than BW-VEH rats (p < 0.005), which did not differ from No-CS groups."

If I am interpreting Figure 1B test data correctly, freezing for the FW-NBQX and FW-VEH group are virtually overlapping for every two-trial block (1-6). Yet the authors report significant differences for NBQX and VEH groups.

Results section: "Fisher's PLSD revealed significant differences in NBQX- and VEH-treated animals in both the BW and FW groups (p's < 0.0005)."

How was this statistical result achieved? If these two groups differ, how could any group fail to differ? Or am I missing something?

3) Experiment 2/Figure 2 is more helpful as a supplement. It is helpful to know backward conditioning is not observed with more limited training. However, even this limited training appeared to produce near ceiling levels of fear to the forward tone. Thus, it was not convincing that a ceiling was avoided to the forward conditioned tone. This is important because it still possible that if weaker fear was acquired during training, then freezing during recall may become sensitive to BNST-NBQX.

4) The authors demonstrate that shock-induced activity, overall, is greater to foot shock on forward trials than on backward trials. In the discussion it is argued that "uncertainty about the timing of the aversive USs increases their aversiveness". Studies of conditioned diminution, and the related concept of conditioned analgesia, would argue exactly the opposite. The observed effect is driven by the predictive ability of the forward cue to diminish the response to foot shock. That is, the backward CS arrangement did not back the US more aversive, instead the forward arrangement made it less aversive.

5) In experiment 3 the authors use a separate manipulation of uncertainty (variable shock intensity) to further probe the role of the BNST. Forward conditioning is performed with fixed or variable shock intensity, followed by a recall test with BNST intact or inactivated. Visually, it appeared that test freezing was greatest in the variable-VEH group, with lesser and equivalent levels of freezing to the remaining three groups. ANOVA found no training condition x drug interaction. However, the n per group is 7-8. Compare this to Figure 1 in which the groups of most interest had 12 (BW-NBQX) and 13 (BW-VEH). The lack of statistical result on generalized contextual freezing would only be convincing in the authors dedicated 12-13 individuals per group.

For Figure 4B, I understand why baseline subtracted freezing was used, but the authors use the same axis label for conditioning, test and test (BL subtraction). That is confusing. Either the bar graphs need their own axis label, or the authors need to report non-baseline subtracted freezing.

Of course, even the variable manipulation does not capture uncertainty. The cue fully predicts foot shock is imminent, just on some trials its more intense than others. The subject is certain the shock will be presented, but uncertain of the specific intensity. Why not use a probabilistic versus deterministic cue?

*Reviewer #3:*

Overall my reaction to this manuscript is somewhat mixed but mostly skewed towards the positive. I do have several questions and comments. I think the manuscript is well written, includes some really nice, sophisticated behaviour, and I enjoyed the thorough analysis of Fos expression in the BNST-projecting afferent structures to determine where information about ambiguous threat might be emerging from. I also enjoyed most of the discussion and thought that some very thoughtful and insightful points were raised there.

However, I do have a couple of major issues that would have to be addressed regarding whether these data do actually warrant the conclusion that the BNST is mediating fear learning to temporally ambiguous stimuli and not to stimuli that are otherwise ambiguous. I will explain these in more detail below.

1) My first question is exactly how ambiguous is a backwards CS using the particular parameters that were employed in this manuscript? Specifically, according to the Materials and methods section, the shock US always precedes the backwards CS, and the intertrial intervals are always 70 seconds. As such, the animal can actually perfectly predict when the shock US will occur using temporal information, including how long it has been since the offset of the backwards CS. If the authors wanted to use a truly temporally ambiguous CS, then would it not make more sense to deliver the shock randomly during the CS presentation? Or alternatively, to randomly alter the length of the intertrial interval?

Another method that could be used to increase the ambiguity of a backwards CS is different intervals between US offset and CS onset, which have been demonstrated to produce excitatory (short intervals 0-4 seconds) versus inhibitory (longer intervals > 10seconds) conditioning, whereas intermediate intervals (4-10seconds) have been shown to produce either type of conditioning (e.g. Sanderson et al., 2014; Hellstern et al., 1998; Delamater et al., 2003). Thus, a backwards CS with a US offset-CS onset interval of 4-10 seconds would perhaps be more 'ambiguous'. Of course, once again if the intertrial intervals are kept consistent at 70 seconds then it would still be questionable whether the US is still "temporally unpredictable."

At minimum, I think an experiment showing the involvement of the BNST in regulating fear to a CS that is truly ambiguous (and not predicted by precise intertrial intervals) is necessary to complete this dataset.

2) There is another sense in which a backwards CS is not really ambiguous, and that is because when the animal receives the backwards CS, they can be sure that they won’t receive the US for at least the duration of that CS and a period of time afterwards. However, on test when the animal does not receive the US prior to CS onset, perhaps the effect of these altered conditions is to introduce some ambiguity (i.e. the shock that yesterday preceded the backwards CS has not yet occurred, therefore it seems possible that it may occur on test). However, the forwards CS also of course suffers this same type of ambiguity when the US does not follow its presentation, such that every presentation after the first might also be rendered ambiguous, and yet BNST inactivation does not affect freezing to the forwards CS. It might be that the types of ambiguity rendered on test to the two types of CSs might manifest differently, but it would be good if the authors could discuss how. Also, more experiments could be needed: BNST affected CS where shock is delivered in a temporally unpredictable manner during CS presentation, and showing a similar effect, OR having an effect on a backwards CS with a US offset-CS onset interval between 4-10 seconds. Finally, a Fos study where animals are sacrificed following a 'test' session during which USs are actually delivered, and the differences in vBNST Fos persist, would provide stronger evidence that it is the ambiguity produced by the different predictiveness of the two types of CSs, rather than the ambiguity produced by the test session.

3) It looks, to me at least, like there might actually be an effect of BNST inactivation in the variable US intensity experiment (results in Figure 4), particularly on the test (4B), although the authors do not report it as such. From the figure, the difference in freezing on test between fixed-veh and variable-veh looks much larger than the difference between fixed-mus and variable-mus. I understand that the interaction is not significant, but the F is high-ish at at 2.7, p >.15 (Results section). Certainly, with the number of animals in this kind of study the authors would have at best a small-medium amount of power to detect a medium effect size. This then begs the question that if more animals were added, would this effect become significant, and most importantly, within the population about which the authors are making inferences, is this an effect? If this were an effect with more animals added, then does it matter? Surely it would simply mean that the BNST is involved in multiple situations of ambiguity of fear predictiveness, rather than just temporally ambiguous stimuli.

The authors subtract baseline freezing to show that (Figure 4B right panel), when higher baseline freezing is taken into account, the rats in fact freeze equally among groups. They justify this by saying the baseline freezing was not significantly different between groups, but the F here really does appear to approach significance (3.6, p >.05, Results section), and as the authors note, freezing looks highest in 'variable-veh' at baseline. This is the group that, having received shocks of variable intensity, would be showing most anxiety (because of the unpredictability of the shock) and thus I do wonder, does it not fit the author's story for BNST-inactivation to be having an effect here?

---

## [Author Response]

[Editors’ note: the author responses to the first round of peer review follow.]

The primary problem, which is mentioned in each review but came out more strongly as an issue requiring further experimentation in the online discussion, is the lack of clear evidence that backward and forward conditioning differ primarily in the uncertainty of the prediction. Each reviewer felt this problem in the manuscript, though for each it is expressed a bit differently. In the end it was felt that addressing this, as well as other concerns about the strength of some of the results, would require more experiments. As eLife has a policy not to invite revisions if they require more than a couple months of experiments, the decision was made to reject. However, the reviewers all agreed that these were potentially very interesting experiments, so if the additional work were done, they were open to the authors coming back, essentially as a resubmission.

Thank you for the summary. These are fair concerns, which we have addressed by adding additional information and experimental data. These items are discussed in each point below and in the revised manuscript. We hope these changes provide significant improvements and merit reconsideration of our manuscript.

Reviewer #1:In the current study, the authors present a series of nicely controlled and straightforward experiments comparing the effects of BNST inactivation on expression of backward and forward conditioned fear responses. […] Overall this provides a very nice set of experiments differentiating the role of this circuit versus amygdalar circuits in these two forms of conditioning.

This a fair overview of our results and we appreciate the positive comments.

What I am less convinced of is that the involvement of the BNST in BW is due to some special role in uncertainty. To some extent this reflects my ignorance regarding BW and how it is conceptualized. In some other studies, procedures such as these are used to examine safety signaling for example. This is not to say I do not see the authors' point; it seems to me that this is something that might be given more explanation in the Introduction (which currently is all about BNST).

We agree that we could have done more to address the conceptualization of backward conditioning and how it might relate to uncertainty. We have expanded the introduction and discussion to better describe the rational and significance of using the forward and backward procedures, and the caveats that are involved in their use. While BW CSs can be used to signal safety, this generally requires a significant number of training trials for discrete CSs. Nonetheless, there is evidence that a BW CS can serve as a conditioned excitor (Chang et al., 2003, and others—see text), which may depend on contextual conditioning and may occur prior to the transition of the signal becoming a conditioned inhibitor. In our hands, the BW training procedure appeared to serve as a conditioned excitor, as evidenced by the increases in freezing we observed following onset of the BW CS and by the lack of increases we see for animals not receiving the CS at test. Regarding uncertainty, we have added an additional experiment which explores the contribution of the BNST to CS that was conditioned using uncertain timing of shock following the offset of the CS. Inactivation of the BNST appeared to mirror the effect of its inactivation during exposure to a backwards CS. We now address in greater detail in the manuscript these outcomes, and how they might explain contributions of the BNST to ambiguous threats.

Perhaps related to this, I am a bit unclear on the significance of the data in Figure 3. This seems to be presented to substantiate that BW is making the subjects fearful but uncertain – is this correct? I wonder if showing this earlier might help support the paradigms. Or showing some direct relationship with the efficacy of BNST inactivation would make this relationship more clear. That is, if a failure to reduce the activity bursts to the US with conditioning reflects uncertainty about the shock and this is mediated by BNST, then the less this reduction, the more BNST should be required for the conditioned responding no? Likewise, it would be useful to know whether there were any relationships within groups between behavior and Fos expression in the various regions.

We have added to the manuscript some additional information that we hope may help better address the finding that activity bursts were higher overall in BW- vs. FW-trained animals. At this time, we have not explicitly addressed whether BNST inactivation reduces the magnitude of activity bursts, but this may be a reasonable expectation. Regarding correlations between behavior and Fos expression, correlations of gross freezing profiles of BW or FW rats did not reveal significant relationships with Fos, suggesting that fear CS exposure was more predictive of Fos expression than freezing performance *per se*. We have noted these findings in the text.

Reviewer #2:[…] The inactivation, tracing and c-Fos imaging methods are all appropriate and are the strength of the manuscript. The innovation comes from the use of backward conditioning. This is a behavioral procedure with a rich history in learning theory but has received little or no attention from neuroscientists. However, no effort is made to draw from this tradition. Instead, the authors immediately cast backwards conditioning as capturing 'threat uncertainty' and 'ambiguous threat' with no convincing explanation. Even more, there are Pavlovian fear conditioning procedures that directly manipulate ambiguity [Rescorla, (1968); McHugh et al., (2015); Erlich, Bush and Ledoux, (2012)]. Thus, there is a strong mismatch between the process the authors set out to investigate, and the processes captured by backward conditioning. The best general advice I can give is for the authors to search for and thoroughly read backward conditioning papers by Heth, Rescorla, Wagner, Ayres, LoLordo, Domjan and their contemporaries.

These are reasonable criticisms, and we have made substantial revisions to better describe what is captured by the backward procedure and our implementation of the work. In particular, we have added more background in the Introduction and have added additional details in the Discussion section.

We have added these and other relevant sources to the text.

Essential revisions:1) In experiment 1 the authors demonstrate the backward conditioning yields less fear, or more extinguishable fear, compared to a forward conditioned cue. Backward fear is also greater in rats given the cue during test, compared to rats given no cue. There is still a concern that rather than producing weak excitatory conditioning, freezing produced by the backward cue is non-associative. That is, equivalent levels of freezing would be observed to a cue unpaired with foot shock. Showing that backward conditioned rats expressed greater freezing in test, compared to unpaired rats, would definitively demonstrate that freezing was the result of associative learning.

This is an important point. Indeed, we do not believe the freezing exhibited by BW CS exposed rats is due (at least not entirely) to non-associative mechanisms, given that the increases in freezing that occur post-baseline are not seen in animals that are tested in the absence of the CS (as mentioned in your comment). That said, we have now added a separate experiment that includes a group of animals that were given footshock US presentations in the absence of any discrete CS; these same rats were later tested in the presence of the novel auditory tone. The results indicated that freezing was not increased after onset of the nonconditioned tone for these animals. We believe this further suggests that there are associative mechanisms involved in the expression of the freezing to the BW-trained CS.

2) ANOVA for Experiment 1 was adequately performed and the main effects & interactions reflect the patterns in the data. However, some issues with post-hoc testing crept up. The following must be a typo, as a group cannot differ from itself:Results section: "Importantly, BW-VEH rats exhibited significantly higher levels of freezing than BW-VEH rats (p < 0.005), which did not differ from No-CS groups."

Indeed, this is a typo, and we have made edits to the manuscript to fix it.

If I am interpreting Figure 1B test data correctly, freezing for the FW-NBQX and FW-VEH group are virtually overlapping for every two-trial block (1-6). Yet the authors report significant differences for NBQX and VEH groups.Results section: "Fisher's PLSD revealed significant differences in NBQX- and VEH-treated animals in both the BW and FW groups (p's < 0.0005)."How was this statistical result achieved? If these two groups differ, how could any group fail to differ? Or am I missing something?

We appreciate the attention to details. This is a mistake on our part; we have updated the text to reflect the significant difference between the BW-NBQX and BW-VEH groups.

3) Experiment 2/Figure 2 is more helpful as a supplement. It is helpful to know backward conditioning is not observed with more limited training. However, even this limited training appeared to produce near ceiling levels of fear to the forward tone. Thus, it was not convincing that a ceiling was avoided to the forward conditioned tone. This is important because it still possible that if weaker fear was acquired during training, then freezing during recall may become sensitive to BNST-NBQX.

We agree the levels of freezing in the FW animals are still relatively high, so we have tempered our interpretation with regards to the number of training trials. We have also moved this Figure in question to the supplemental section. That said, the levels of fear elicited by the 5-trial FW animals may be considered similar to that of the 12-trial BW animals from Figure 1 – with only the 12-trial BW animals being sensitive to the BNST inactivation. Nevertheless, we have not explicitly tested whether 1-trial training is sensitive to the BNST inactivation.

4) The authors demonstrate that shock-induced activity, overall, is greater to foot shock on forward trials than on backward trials. In the discussion it is argued that "uncertainty about the timing of the aversive USs increases their aversiveness". Studies of conditioned diminution, and the related concept of conditioned analgesia, would argue exactly the opposite. The observed effect is driven by the predictive ability of the forward cue to diminish the response to foot shock. That is, the backward CS arrangement did not back the US more aversive, instead the forward arrangement made it less aversive.

This is a fair point, and may be poor wording on our part, as we were intending to suggest that the US (in the BW example) may be more aversive relative to the FW CS due to the decrease in the FW paradigm. Regardless, you bring up an important point, and we have edited the text to clarify these items.

5) In experiment 3 the authors use a separate manipulation of uncertainty (variable shock intensity) to further probe the role of the BNST. Forward conditioning is performed with fixed or variable shock intensity, followed by a recall test with BNST intact or inactivated. Visually, it appeared that test freezing was greatest in the variable-VEH group, with lesser and equivalent levels of freezing to the remaining three groups. ANOVA found no training condition x drug interaction. However, the n per group is 7-8. Compare this to Figure 1 in which the groups of most interest had 12 (BW-NBQX) and 13 (BW-VEH). The lack of statistical result on generalized contextual freezing would only be convincing in the authors dedicated 12-13 individuals per group.

We agree that more animals would increase the power of this experiment (though 7-8 animals per group is not uncommon for the field). Nonetheless, CS-elicited freezing is still substantial in all groups (as indicated by the baseline subtraction). In other words, the freezing in the presence of the CS is only increased relative to the baseline and is not masked by it. Thus, and with more animals, the BNST inactivation may reduce contextual generalization for the variable animals, but it does appear that this would prevent the freezing elicited by the CS itself (as freezing levels are equivalent in the baseline subtraction and is similar in both drug groups). We have elaborated on these items in the manuscript.

For Figure 4B, I understand why baseline subtracted freezing was used, but the authors use the same axis label for conditioning, test and test (BL subtraction). That is confusing. Either the bar graphs need their own axis label, or the authors need to report non-baseline subtracted freezing.

We have edited this Figure to show the freezing values before baseline subtraction, and after, with the subtraction separated from the other parts of the Figure. We hope this improves the clarity.

Of course, even the variable manipulation does not capture uncertainty. The cue fully predicts foot shock is imminent, just on some trials its more intense than others. The subject is certain the shock will be presented, but uncertain of the specific intensity. Why not use a probabilistic versus deterministic cue?

We have modified the language used here to address the concept of uncertainty. Indeed, our argument is that both the fixed and variable US magnitude groups are trained with a sense of temporal certainty (i.e., the shock is coming soon), but differ in the expectation of the magnitude of the outcome. We wished to include these data as it may help to address whether the BNST contributes to learning about cues that predict variable aversive outcomes (which may be distinct from temporal uncertainty). While we did not address probabilistic cues in this manuscript (as we wanted to equate US exposure, in particular), we have added an additional experiment which includes random timing of shock, which may help with the overall interpretation of the data.

Reviewer #3:[…] However, I do have a couple of major issues that would have to be addressed regarding whether these data do actually warrant the conclusion that the BNST is mediating fear learning to temporally ambiguous stimuli and not to stimuli that are otherwise ambiguous. I will explain these in more detail below.1) My first question is exactly how ambiguous is a backwards CS using the particular parameters that were employed in this manuscript? Specifically, according to the Materials and methods section, the shock US always precedes the backwards CS, and the intertrial intervals are always 70 seconds. As such, the animal can actually perfectly predict when the shock US will occur using temporal information, including how long it has been since the offset of the backwards CS. If the authors wanted to use a truly temporally ambiguous CS, then would it not make more sense to deliver the shock randomly during the CS presentation? Or alternatively, to randomly alter the length of the intertrial interval?

Indeed, this is a fair point with regards to the issue of timing. Indeed, it is possible that the animals could be using the static intervals to be timing the onset of shock (the same could be said for the FW groups, which are separated by the same duration of intervals). Nevertheless, we have added an additional experiment that explicitly modulates the timing of the shock, relative to the CS. Specifically, animals received presentations of the CS with randomized onset of shock following CS offset. BNST inactivation appeared to reduce freezing to a similar extent as the BW CS, suggesting similar roles for the BNST in both. That said, we have modified the text to better clarify the interpretations of our data set.

Another method that could be used to increase the ambiguity of a backwards CS is different intervals between US offset and CS onset, which have been demonstrated to produce excitatory (short intervals 0-4 seconds) versus inhibitory (longer intervals > 10seconds) conditioning, whereas intermediate intervals (4-10seconds) have been shown to produce either type of conditioning (e.g. Sanderson et al., 2014; Hellstern et al., 1998; Delamater et al., 2003). Thus, a backwards CS with a US offset-CS onset interval of 4-10 seconds would perhaps be more 'ambiguous'. Of course, once again if the intertrial intervals are kept consistent at 70 seconds then it would still be questionable whether the US is still "temporally unpredictable."At minimum, I think an experiment showing the involvement of the BNST in regulating fear to a CS that is truly ambiguous (and not predicted by precise intertrial intervals) is necessary to complete this dataset.

These are important points and important studies. Indeed, and as mentioned in the previous response, we have included an experiment where we modulated the intervals between CS offset and US onset at each trial (impeding the ability the animals to properly time the onset of the US). As was similar for BNST inactivation in the presence of a BW CS, we found BNST inactivation reduced freezing to the randomized CS. We have edited the text to incorporate these new data.

2) There is another sense in which a backwards CS is not really ambiguous, and that is because when the animal receives the backwards CS, they can be sure that they won’t receive the US for at least the duration of that CS and a period of time afterwards. However, on test when the animal does not receive the US prior to CS onset, perhaps the effect of these altered conditions is to introduce some ambiguity (i.e. the shock that yesterday preceded the backwards CS has not yet occurred, therefore it seems possible that it may occur on test). However, the forwards CS also of course suffers this same type of ambiguity when the US does not follow its presentation, such that every presentation after the first might also be rendered ambiguous, and yet BNST inactivation does not affect freezing to the forwards CS. It might be that the types of ambiguity rendered on test to the two types of CSs might manifest differently, but it would be good if the authors could discuss how. Also, more experiments could be needed: BNST affected CS where shock is delivered in a temporally unpredictable manner during CS presentation, and showing a similar effect, OR having an effect on a backwards CS with a US offset-CS onset interval between 4-10 seconds. Finally, a Fos study where animals are sacrificed following a 'test' session during which USs are actually delivered, and the differences in vBNST Fos persist, would provide stronger evidence that it is the ambiguity produced by the different predictiveness of the two types of CSs, rather than the ambiguity produced by the test session.

These are fair points. As per the other responses, we have included new data that directly modulates timing of shock onset, which may best address the possibility of BNST contributions to cues of uncertain timing of shock (see above). However, in the current data set, we cannot remove the possibility that the absence of the US at the test doesn’t contribute to a sense of ambiguity that is sensitive to BNST inactivation (but, as mentioned, this would be true for both the FW and BW CS, in which BNST inactivation was selective only to the BW CS). Additionally, the presence of shocks at test would be somewhat confounded by the new learning that would take place as a result. We do not have Fos data to speak to contributions of the BNST at the time of a retrieval session that includes the presence of shocks. That said, we have cited studies that have found Fos in the BNST after shock exposure generally, so it may be difficult to isolate recall-induced Fos in the BNST in the presence of shocks from new learning-induced Fos activation in the BNST (which is why, in part, we tested animals in a shock-free state).

3) It looks, to me at least, like there might actually be an effect of BNST inactivation in the variable US intensity experiment (results in Figure 4), particularly on the test (4B), although the authors do not report it as such. From the figure, the difference in freezing on test between fixed-veh and variable-veh looks much larger than the difference between fixed-mus and variable-mus. I understand that the interaction is not significant, but the F is high-ish at at 2.7, p >.15 (Results section). Certainly, with the number of animals in this kind of study the authors would have at best a small-medium amount of power to detect a medium effect size. This then begs the question that if more animals were added, would this effect become significant, and most importantly, within the population about which the authors are making inferences, is this an effect? If this were an effect with more animals added, then does it matter? Surely it would simply mean that the BNST is involved in multiple situations of ambiguity of fear predictiveness, rather than just temporally ambiguous stimuli.

Similar to what we discussed above, it is possible that the null results are due to the lower number of animals compared to the other experiments. However, the potential effect seems to be driven by differences in the baseline freezing, which is why we include the baseline subtraction data. In other words, the effect we are seeing is the result of BNST inactivation on generalized context fear (or sensitized responding more generally), but not CS-elicited fear (as it is riding on top of the baseline). Subtracting the baseline does not eliminate the significant increase in CS-elicited freezing (regardless of drug treatment). Conversely, BNST inactivation eliminated CS-elicited freezing to the 12-trial BW CS (and for the RANDOM CS), regardless of the baseline.

The authors subtract baseline freezing to show that (Figure 4B right panel), when higher baseline freezing is taken into account, the rats in fact freeze equally among groups. They justify this by saying the baseline freezing was not significantly different between groups, but the F here really does appear to approach significance (3.6, p >.05, Results section), and as the authors note, freezing looks highest in 'variable-veh' at baseline. This is the group that, having received shocks of variable intensity, would be showing most anxiety (because of the unpredictability of the shock) and thus I do wonder, does it not fit the author's story for BNST-inactivation to be having an effect here?

We have clarified these points in the text (also, see responses above). Indeed, we believe the heightened freezing at baseline is consistent with a role for the BNST in generalized context fear (which may involve some BNST-dependent sensitization), but nevertheless, these potential impacts of BNST inactivation on freezing are specific to the baseline (but not the CS-elicited freezing, as indicated by the increase in freezing regardless of baseline levels), suggesting that CS-elicited freezing to the variable FW CS is not dependent on the BNST for expression.